# Fast light-switchable polymeric carbon nitride membranes for tunable gas separation

Timur Ashirov [1], Julya Stein Siena [2], Mengru Zhang[3], A. Ozgur Yazaydin [3], Markus Antonietti [2] ✉ & Ali Coskun [1] ✉

Switchable gas separation membranes are intriguing systems for regulating the transport properties of gases. However, existing stimuli-responsive gas separation membranes suffer from either very slow response times or require high energy input for switching to occur. Accordingly, herein, we introduced light-switchable polymeric carbon nitride (pCN) gas separation membranes with fast response times prepared from melamine precursor through in-situ formation and deposition of pCN onto a porous support using chemical vapor deposition. Our systematic analysis revealed that the gas transport behavior upon light irradiation is fully governed by the polarizability of the permeating gas and its interaction with the charged pCN surface, and can be easily tuned either by controlling the power of the light and/or the duration of irradiation. We also demonstrated that gases with higher polarizabilities such as $CO_2$ can be separated from gases with lower polarizability like $H_2$ and He effectively with more than 22% increase in the gas/$CO_2$ selectivity upon light irradiation. The membranes also exhibited fast response times (<1 s) and can be turned "on" and "off" using a single light source at 550 nm.

Ever increasing greenhouse emissions, water and air pollution have promoted the development of functional membranes with tailored properties for the desired applications. Dynamic membranes with switchable properties are on the other hand rather interesting as they can adapt to their environment by means of an external stimulus. Particularly, the idea of responsive membranes that resembles biomimetic behavior similar to cell membranes has been a topic of great interest[1,2]. The physical and chemical properties of these membranes can be changed (switched) under the influence of external stimuli such as temperature, pH, light, electric and magnetic field, ionic strength etc. The realization of such membranes could enable the development of so-called "smart membranes"[3,4]. Indeed, responsive membranes have already been used in drug delivery applications, wherein the membranes can be activated by an external stimuli such as pH, temperature and ionic strength[5]. However, for dynamic applications such

as gas separation, membranes that can be switched by light[6,7], electric[8] and magnetic field are considered to be more suitable[9]. Recently, Knebel and coworkers reported an electric field-switchable metal-organic framework (MOF) membrane bearing zeolitic imidazolate framework, ZIF-8, which transforms into a polymorph structure under the influence of an electric field, thus resulting in a hindered gas transport[8]. Switching of the membrane towards Cm polymorph was fast, however, relaxation of the polarized ZIF-8 required ~1.5 h, which can be potentially improved by applying additional stimulus (e.g. temperature)[8]. On the other hand, light-responsive membranes featuring switchable molecular components such as azobenzenes and spyropyrans require less energy for switching[10,11]. For instance, Zhu and coworkers showed tunable $CO_2$ uptake in porous organic polymers functionalized with azobenzene moieties, where after UV irradiation $CO_2$ uptake capacity has increase up to 29%[12]. They explained the

[1]Department of Chemistry, University of Fribourg, Chemin du Musee 9, 1700 Fribourg, Switzerland. [2]Department of Colloid Chemistry, Max Planck Institute of Colloids and Interfaces, Am Mühlenberg 1, D-14476 Potsdam, Germany. [3]Department of Chemical Engineering, University College London, Torrington Place, London WC1E 7JE, UK. ✉e-mail: office.cc@mpikg.mpg.de; ali.coskun@unifr.ch

increase in $CO_2$ binding capacity is arising due to difference in polarities of *cis* and *trans* azobenzene[12]. One of the easiest ways of preparing light-responsive membranes is to incorporate light-switchable moieties into a polymer matrix. First example of such photo-switchable membranes used to study tunable ion transport[13]. Later, Weh and coworkers embedded azobenzenes into the poly(-methylmethacrylate) (PMMA) and observed decrease in the permeance of $CH_4$, $H_2$, $SF_6$ etc. upon photoswitching[14]. Other examples also include liquid crystal as host materials for azobenzenes[15], where switchable nitrogen permeation was studied under constant pressure and volume[16]. In another example, Bujak and coworkers synthesized azopolyimides and studied switchable permeation of He, $N_2$, $O_2$ and $CO_2$[17]. They observed 1.64% change in the permeation of He, however, the $CO_2$ permeance decreased by 7.89% due to dipole-quadrupole interactions. Recently, Nocon-Szmajda copolymerized and molecularly dispersed azobenzenes in polyimide matrix and observed almost ~28% and ~22% decrease in the $CO_2/N_2$ selectivity in the case of molecular dispersion and copolymerization, respectively, upon 5 min irradiation[18]. However, the permeance values of these membranes were extremely low in the ranges of ~$10^{-2}$–$10^{-3}$ GPU.

MOFs are also interesting materials for light-switchable gas separation applications as they can offer higher permeances. For instance, Prasetya and coworkers prepared light-responsive mixed-matrix membranes by using JUC-62 and PCN-250 as filler materials and matrimid as polymer matrix[19]. They observed up to 8.5% decrease in $CO_2$ permeance upon irradiation at 15 wt.% JUC-62 loading, however, ~83 min of irradiation was required to observe this change. In another example, Knebel and coworkers incorporated azobenzene guest molecules into an ultrathin UiO-67 membrane and used light to control the gas permeation and selectivity. The authors observed a decrease in the $H_2/CO_2$ separation factor from 14.7 to 10.1, however, the switching azobenzenes into *cis* and back to *trans* configurations required rather long irradiation times 120 and 106 min, respectively[20]. In another study, Wang and coworkers prepared MOFs embedding azobenzene moieties and observed two-times and 1.5-times increase in the $H_2/CO_2$

and $N_2/CO_2$ selectivities upon light irradiation, respectively[7]. Additionally, the switching times also improved to ~15 min for *cis* and ~5 min back to *trans*. These switching times are still rather long owing to the rigid membrane structure restricting the conformational changes of switchable components[10,21,22].

On the other hand, the switching event without involving conformational changes of molecular components can offer faster response times[1]. One such material is the polymeric carbon nitride (pCN), in which the switching occurs through the photo-induced separation of electrons and holes[23–25]. Polymeric carbon nitride is part of a class of binary compounds, named carbon nitrides. It is synthesized from nitrogen-rich organic compounds such as urea and melamine through a thermal polymerization process[26]. Recent studies on the polymerization mechanism that leads to the final 2D structure have shown that pCN is formed by linear linked heptazines connected by hydrogen bonds with an offset, e.g. the 2D sheets are not precisely aligned on the top of each other, which causes the molecule to bend[27,28]. Thus, the structure of polymeric carbon nitride would strongly depend on deposition parameters and final carbon to nitrogen ratio (Fig. 1). Upon illumination of pCN, the electrons are separated from the holes and moved to the bulk carbon nitride, thus creating an asymmetric surface charge distribution[29,30]. Separation of electrons and holes happens on a very different timescale compared to molecular isomeric light response. Due to its light responsiveness, pCN has already found several applications in energy conversion[23], catalysis[24], gas separation[31,32], and artificial photosynthesis[30], etc. Although it has been theoretically predicted that polymeric carbon nitride could show very high $H_2$/gas and He/gas selectivities[33–36], the light-switchable gas transport in pCN membrane has not been investigated before. In this direction, herein, we have studied the gas transport behavior and the effect of light switching in the pCN membranes. The membranes were prepared via in-situ formation and deposition of pCN onto an anodic aluminum oxide (AAO) membrane via low-pressure chemical vapor deposition (LPCVD) using melamine as an organic precursor[37,38]. In order to control the membrane thickness and porosity, we

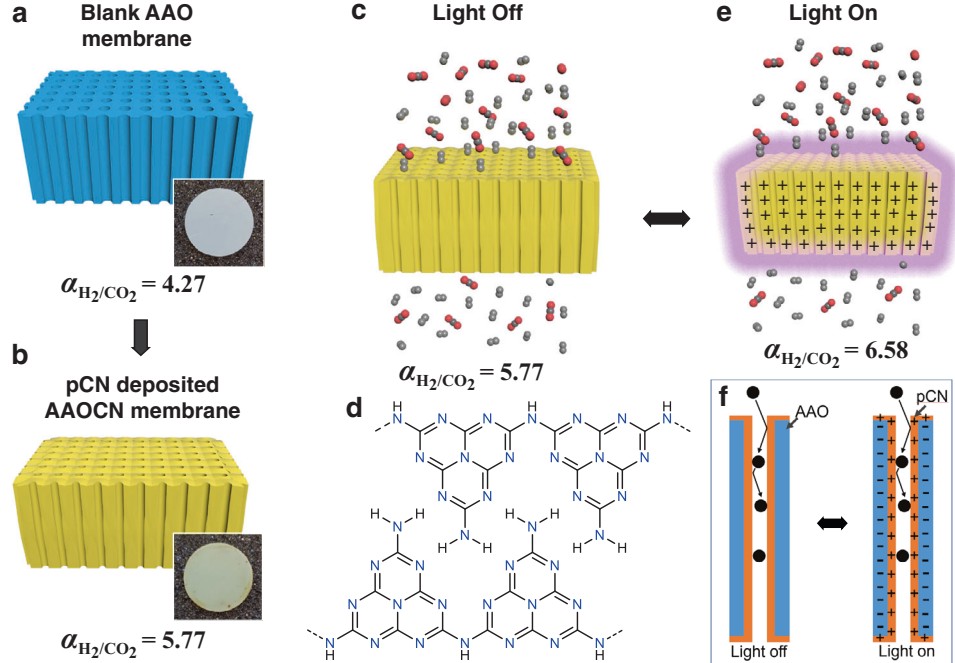

**Fig. 1 | Schematic representations of pCN membrane preparation.** Cross-sectional image of (**a**) blank AAO membrane (Inset: real picture of AAO membrane), (**b**) polymeric carbon nitride deposited AAOCN membrane (Inset: real picture of AAOCN membrane), (**c**) gas transport through AAOCN membrane without irradiation, (**d**) molecular structure of pCN (**e**) formation of a charged pCN surface upon light irradiation and its effect on gas transport through the membranes. **f** schematic side-view of formation of charges on pCN upon irradiation.

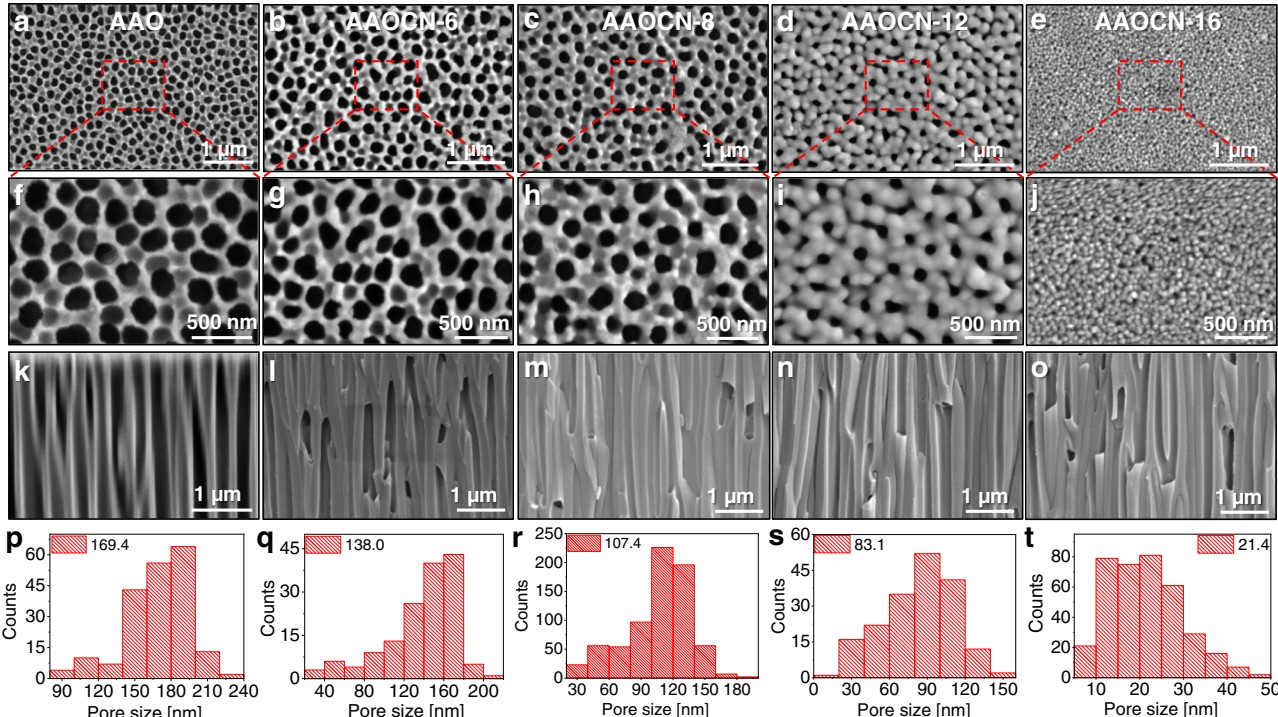

**Fig. 2 | Electron microscopy characterizations of pCN membranes. a–e** Top-view, **f–j** zoomed-in regional, and **k–o** cross-sectional view SEM images of blank AAO and AAOCN-X membranes, and **p–t** pore size distribution histograms of the SEM images in **a–e**.

systematically varied the amount of organic precursor, melamine, and correlated with the gas transport characteristics. We observed a clear increase in the $H_2/CO_2$ and $He/CO_2$ selectivities upon deposition of pCN surpassing the Knudsen selectivity (Fig. 1a, b). The switching of the membrane occurred almost immediately in less than a second upon light irradiation at 550 nm with a more than ~22% increase in the gas/$CO_2$ selectivity (Fig. 1c–f). Moreover, the relaxation of membranes occurred instantly upon stopping light irradiation. Notably, we demonstrated a clear correlation between the gas polarizability and the interaction with the charged pCN surface upon light irradiation, thus offering a responsive membrane for the separation of gases by means of their polarizability.

## Results

### Preparation of the pCN membranes

Polymeric carbon nitride was deposited by following a previously reported approach and using a readily available and cheap organic precursor, melamine (Supplementary Table 1)[37,38]. We identified commercial AAO membranes with an average pore size of 100 nm as a suitable porous substrate for the deposition of pCN due to their low-cost and scalability. In order to identify the ideal membrane thickness and porosity, we deposited pCN membranes starting from 6, 8, 12, 16 and 20 g of melamine (sample name AAOCN-X where X is the amount of melamine used). The successful deposition of pCN onto the AAO membranes was verified by Fourier-transform infrared (FTIR) spectroscopy (Supplementary Fig. 1) and scanning electron microscopy (SEM) (Fig. 2a–c) analyses. The FTIR spectra of the AAOCN-16 sample revealed the characteristic carbon nitride stretching bands of -C-N- at ~1233, ~1406, -C = N- at ~1556, 1619 cm⁻¹ and -C-N- bending band at ~805 respectively[39,40]. Moreover, the molecular structure of pCN was analyzed using X-ray photoelectron spectroscopy (XPS). The XPS survey spectra revealed ~36% of carbon, ~50% of nitrogen and ~14% of oxygen content (Supplementary Fig. 2). High resolution C $1s$ spectra on the other hand showed the presence of -C-C-/-C = C- and −C-N-/C = N- moieties at 285.5 and 288.9 eV, respectively (Supplementary

Fig. 3a)[23,26,41]. Moreover, high resolution N $1s$ spectra revealed the presence of peaks at 399.7 and 400.6 eV that were attributed to typical -C-N- and -C = N- bonds of carbon nitride, respectively (Supplementary Fig. 3b)[23,26,41]. Most importantly, the peaks observed on high resolution O $1s$ spectra were attributed to AAO moieties and organic oxygen moieties were not detected (Supplementary Fig. 3c)[42]. We also measured the ultraviolet-visible (UV-Vis) spectra of AAOCN and blank AAO membranes (Supplementary Fig. 4). pCN showed a rather broad adsorption peak ranging from 250 to 500 nm, which is in good agreement with the previously reported pCN[43], while blank AAO absorbs deep in UV region below 250 nm. The UV-Vis absorption bands of pCN also match with emission spectra of used LED light source (Supplementary Fig. 5).

During the preparation of the membranes, we realized that the position of the sample in the LPCVD chamber has an enormous effect on the porosity of the membrane, e.g., placing the sample too far from the source leads to a surface growth and a subsequent pore blockage (Supplementary Fig. 6). We hypothesize that this phenomenon is due to changes both in the reactant flow and temperature profile along the hot-wall tubular reactor of the low-pressure CVD system. When locating the substrate further away from the precursor, e.g., closer to the end of the furnace, gas-nucleation can occur, leading to local depletion of the reactant and non-homogeneous thickness. A change in the mobility-diffusion of the reactants on the surface of the AAO membrane can result in the overgrowth of neighboring grains, causing the pore surface to clog and voids within it.

The top-view and cross-sectional SEM images in Fig. 2 confirmed the successful deposition of pCN onto the substrate. The pore size of blank AAO membrane ranging between 90 and 240 nm with an average pore size of 169.4 nm (Fig. 2a–j) started to narrow down upon deposition of pCN, and the pore size decreased almost linearly with the precursor amount (Fig. 3a), eventually being completely blocked at the deposition of 20 g of melamine (Supplementary Fig. 7). The growth of pCN inside the pores was also verified by the cross-sectional SEM images in Fig. 2k–o. Additionally, pCN deposition was also confirmed

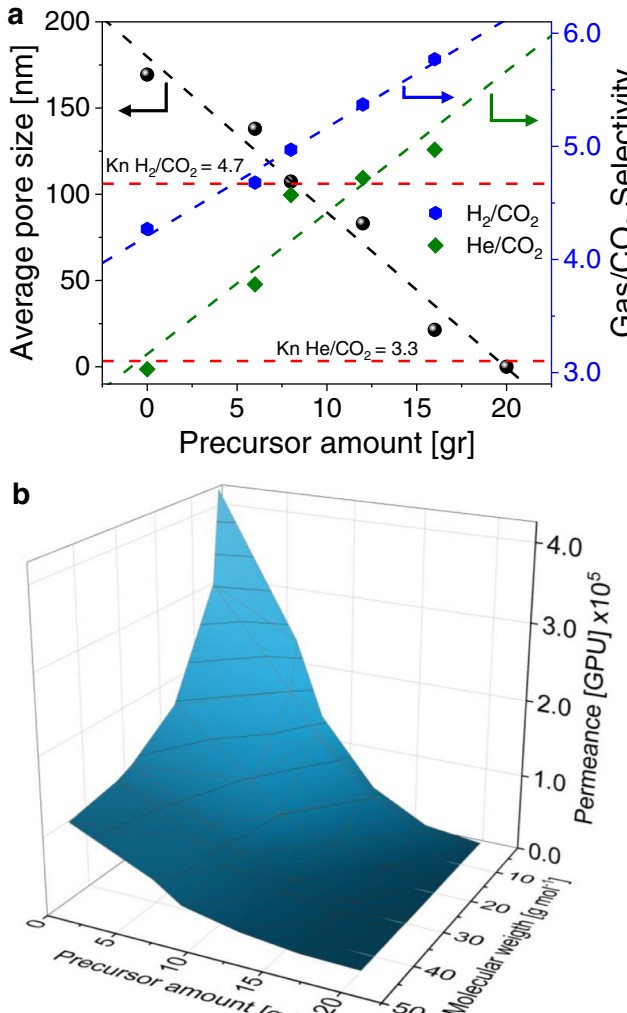

**Fig. 3 | Gas permeation results of AAOCN membranes. a** Average pore size, $H_2/CO_2$ and $He/CO_2$ selectivities vs the precursor amount. **b** The gas permeance vs precursor amount and molecular weight of the gas.

visually from the color change of the bare AAO membrane from white to pale yellow, which is the characteristic color of carbon nitrides (Fig. 1a, b). This observation points to a relatively high growth rate of the films and the related cluster growth of coating material in the gas phase, i.e., we move from molecular deposition to a patchier deposition along the device.

## Gas transport through the pCN membranes

In order to understand the gas transport properties of AAOCN membranes, we have studied six different gases: $H_2$, $He$, $CH_4$, $N_2$, $O_2$ and $CO_2$ (Supplementary Fig. 8). Separation in blank AAO happens due to differences in the masses of gases (Knudsen diffusion) due to its large pores with a tubular shape (~60 μm). The initial $H_2/CO_2$ selectivity was 4.27, (Fig. 3a) with a permeance of $4.2 \times 10^5$ GPU (1 GPU = $3.35 \times 10^{-10}$ mol$^{-1}$ s$^{-1}$ m$^{-2}$ Pa$^{-1}$). However, upon deposition of pCN, the $H_2/CO_2$ selectivity started to increase linearly, eventually surpassing the Knudsen selectivity (~4.69) and reaching to 5.77 for the AAOCN-16 membrane at $H_2$ permeance of $1.3 \times 10^4$ GPU. On the other hand, the change in the $He/CO_2$ selectivity poorly fitted the linear model. Despite the decrease in the pore size upon pCN deposition, the pores were still large for molecular sieving to take place, thus the surface diffusion mechanism started to dominate due to the presence of long channels (~60 μm)[44]. Increasing the amount

of deposited pCN led to both pore size reduction and stronger gas-pore wall surface interactions. Due to the higher polarizability of $CO_2$ compared to $H_2$ and $He$[45], it interacts strongly with pCN surface, thus leading to an increase in the $H_2/CO_2$ and $He/CO_2$ selectivities (Fig. 3a)[46]. Initially, permeance vs molecular weight curve fitted with the Knudsen model, where the permeance was correlated with the square root of the molecular weight[44,47]. However, after deposition of pCN, it started to deviate from the Knudsen model (Supplementary Fig. 9) and the permeance decreased exponentially, going eventually down to zero for the AAOCN-20 membrane (Fig. 3b). Figure 3b shows the relation of gas permeance with the precursor amount (grams) and molecular weight (g mol$^{-1}$) of the gases. The gas permeance decreased with respect to the precursor amount, and heavier gases showed lower permeance.

## Light-switchability of pCN membranes

It is well-known that the switching mechanism on pCN occurs via the separation of electrons and holes upon light irradiation[23,25,32]. Carbon nitride membranes on various supports have been investigated previously for active transport of cations in an aqueous media, wherein Antonietti and coworkers demonstrated that the presence of an asymmetric surface charge distribution of the carbon nitride under light irradiation[48,49]. This phenomenon could lead to enhanced gas-pore wall surface interactions[50], thus affecting the gas permeation behavior (Fig. 1c–f). In order to understand the effect of light-switching of pCN membrane on the transport properties of gases, different light power levels in the range of 0–7.2 W/cm$^2$ and their effect on the gas permeation and selectivity were systematically investigated. Initial light power level tests of AAOCN-6 at 250 mbar transmembrane pressure showed that the permeance of all gases decreased with an increasing power level owing to the enhanced gas-surface interactions (Supplementary Figs. 10-11a–c)[46]. However, we noticed the change in the permeance was not the same for all gases. Whereas the lowest reduction in permeance at maximum light power was observed for $H_2$ and $He$, the highest one was realized for $CO_2$, which was attributed to the higher polarizability of $CO_2$ compared to the $H_2$ and $He$ (Supplementary Table 2). Similarly, due to the high polarizability of $CO_2$, the gas/$CO_2$ selectivity also increased with the light power level (Supplementary Figs. 10a–c and 11a–c). AAOCN-8 membrane showed higher increase in the gas/$CO_2$ selectivity at maximum power of 7.2 W/cm$^2$, compared to AAOCN-6 primarily due to the pore narrowing, except for $CH_4/CO_2$ selectivity, owing to the high polarizability of both gases (Supplementary Figs. 12–13). Similarly, the decrease in the permeance and increase in the gas/$CO_2$ selectivity at 7.2 W/cm$^2$ further enhanced on AAOCN-12 (Supplementary Figs. 14–15a–c). Finally, there was ~15% decrease in the permeance of $CO_2$ on AAOCN-16 membrane, which in return led to ~11% increase in the $H_2/CO_2$ and $He/CO_2$ selectivities at 250 mbar transmembrane pressure (Fig. 4a–c). Even the $CH_4/CO_2$ selectivity increased by ~4.5% despite having very similar polarizabilities (Supplementary Figs. 16a–c). In order to understand the correlation between the permeance decrease upon irradiation at maximum light power intensity of 7.2 W/cm$^2$, amount of deposited pCN and gas polarizability, the permeance decrease was plotted with respect to gas polarizability for all the membranes at different precursor amounts (Fig. 5a). The decrease in the permeance for gases with low polarizability, such as $He$ and $H_2$, did not have a strong correlation with the amount of deposited pCN, as these gases do not interact very strongly with the pCN surface upon light irradiation. On the other hand, the reduction in the permeance increased drastically with the amount of deposited pCN for gases with high polarizability, that is $CO_2$ and $CH_4$ (Fig. 5a).

The growth of pCN on the surface of the AAO leads to the formation of a heterojunction, in which the photoexcited electrons in the lowest unoccupied molecular orbital (LUMO) of pCN can migrate to

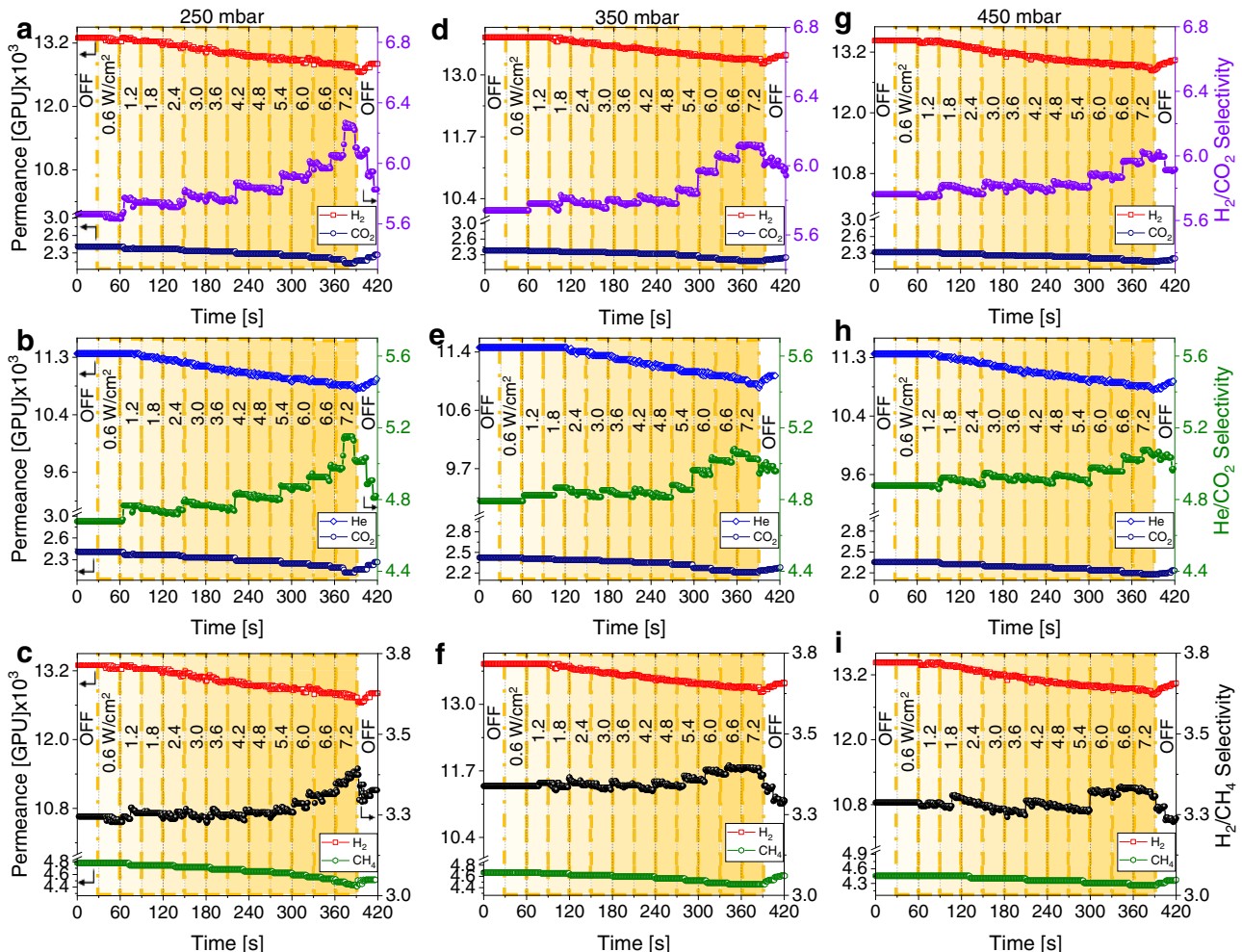

**Fig. 4 | Light switchability of AAOCN membranes at different irradiation power levels.** The change of gas permeance and selectivity vs light intensity (W/cm$^2$) on **AAOCN-16** for $H_2/CO_2$, $He/CO_2$ and $H_2/CH_4$ separations at 250 mbar (**a–c**), at 350 mbar (**d–f**) and at 450 mbar (**g–i**) transmembrane pressure, respectively. The light intensity (W/cm$^2$) is shown in the middle of each plot and increases from left to right in accordance with the intensity of yellow background.

the defect sites of AAO as it was also demonstrated by Li et al[51]. for g-C$_3$N$_4$/Al$_2$O$_3$ heterojunctions, where they observed an improved charge transfer. In order to understand the affinities of $CO_2$, $CH_4$, $H_2$, and He towards pCN, their binding energies were estimated by carrying out periodic plane-wave DFT calculations as implemented in the CASTEP program (version 19.11)[52]. In these periodic DFT calculations, we used a heptazine based graphitic carbon nitride (g-C$_3$N$_4$) structural model reported by Wang et.al[53], where the structure was determined through ab initio evolutionary search and consequent experimental verification. The systems were described using a simulation cell containing 2 layers of g-C$_3$N$_4$, obtained by replicating the unit cell 2 × 2, and 20 Å of vacuum, which was placed to avoid the interaction between two periodic images along the coordinate perpendicular to the g-C$_3$N$_4$ layers and a single gas molecule (Supplementary Fig. 17). Binding energies were calculated for both ground state and excited systems (Supplementary Table 3). For the latter, a core hole is assigned to one of the nitrogen atoms on the top layer; i.e. the layer exposed to the gas molecule, by exciting one of the 1s$^2$ electrons of the selected nitrogen atom, which resulted in an overall +1 charge on the top carbon nitride layer. As expected, in the ground state of g-C$_3$N$_4$ layers, we observed the highest binding energy for $CO_2$ followed by $CH_4$, which is also in agreement with the observed gas selectivities. When a core hole is created, we observed an increase in the dipole moment of g-C$_3$N$_4$ layers from 4.53 debye in the ground state to 4.95 debye in the charged

state. Moreover, the calculation of binding energies of gases for the +1 charged state revealed that the binding energy of both $CO_2$ and $CH_4$ increased, which is expected owing to their higher polarizability, whereas rather modest changes observed for $H_2$ and He. These results explain the change in the gas selectivity of pCN membrane upon light irradiation and its correlation with the polarizability of gas molecules as it is expected that the higher the polarizability of the molecule the greater its diffusion will slow down near the pore surface due to increased electrostatic interactions after light irradiation.

**The effect of transmembrane pressure**

Another factor which plays an important role in the stimuli-responsive membranes is the transmembrane pressure as the surface interactions depend highly on the gas flow rate through the membrane[1,4,10,50]. Initially, we carried out the light power tests at 250 mbar transmembrane pressure. Afterwards, we also tested 350 (Fig. 4d–f and Supplementary Figs. 10–16d–f) and 450 mbar transmembrane pressures (Fig. 4g–i and Supplementary Figs. 10–16g–i) to understand the effect of pressure. Increasing the transmembrane pressure led to a faster gas permeation and less gas-pore wall interactions, thus leading to a lower permeance decrease upon irradiation, especially for the gases with higher polarizabilities (Supplementary Fig. 18). This effect was not noticeable (<2% change) in the AAOCN-6 and AAOCN-8 membranes due to the presence of large pores and low amount of deposited pCN

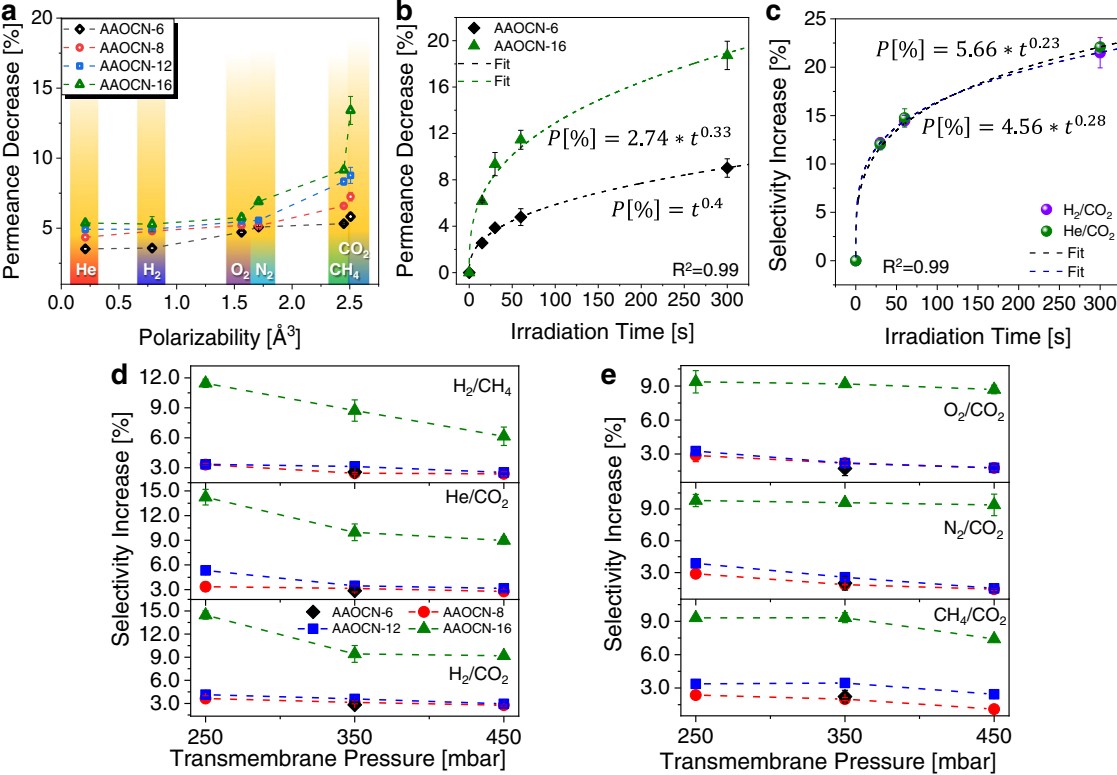

**Fig. 5 | Light switchable gas transport behavior of AAOCN membranes at different irradiation times and transmembrane pressures. a** The permeance decrease upon irradiation vs gas polarizability in AAOCN-X membranes at 250 mbar transmembrane pressure. **b** The $CO_2$ permeance decrease (%) upon irradiation vs irradiation time (s) on AAOCN-6 and AAOCN-16 membranes. **c** The increase in the $H_2/CO_2$ and $He/CO_2$ selectivities upon irradiation vs the irradiation time on AAOCN-16 membrane at 250 mbar transmembrane pressure. **d** The increase in the gas selectivity over 60 s irradiation at maximum light intensity of 7.2 W/cm² vs transmembrane pressure for $H_2/CO_2$, $He/CO_2$, $H_2/CH_4$ mixtures and **e** for $CH_4/CO_2$, $N_2/CO_2$, $O_2/CO_2$ on AAOCN-X membranes. The error bars are generated from the standard deviations of measurements using various number of samples.

in the pores (Supplementary Figs. 10–13d–i and 18). However, the effect of pressure on both gas permeance and selectivity became obvious for AAOCN-12 (Supplementary Figs. 13–14d–i and 18) and AAOCN-16 (Fig. 4d–i and Supplementary Figs. 16d–i, 18) membranes owing to their smaller pore sizes. The permeance decrease upon irradiation at higher pressures for AAOCN-12 membrane led to 2-3% change in the gas/$CO_2$ selectivity. On the other hand, the increase in the gas/$CO_2$ selectivity at 7.2 W/cm² light power level decreased gradually at higher pressures for AAOCN-16 membrane, e.g., while 11% increase in the $H_2/CO_2$ selectivity was observed at 250 mbar, this increase was recorded as ~7 and 5% at 350 and 450 mbar, respectively (Fig. 4d–i). Similar reduction in the gas/$CO_2$ selectivities at higher transmembrane pressures were also observed for other gas/$CO_2$ mixtures (Supplementary Fig. 16d–i). Since the highest gas selectivity was obtained at a maximum power level of 7.2 W/cm², subsequent experiments were carried out at this light power level. It should be noted that increasing the light power level is expected to lead to a higher increase in the gas/$CO_2$ selectivity.

**The effect of the irradiation duration**
Following the light power tests, we also investigated the effect of light irradiation time. We tested different irradiation times and discovered that the gas permeance decrease was proportional with the irradiation time (Supplementary Fig. 19). The response time of the membranes turned out to be extremely fast within seconds and changes occurred instantly (<1 s). In order to understand the effect of irradiation time, we plotted the irradiation time vs $CO_2$ permeance decrease for AAOCN-6 and AAOCN-16 membranes, which were chosen to clearly demonstrate the effect of pore size and pCN amount (Fig. 5b). Both curves showed a

logarithmic relation rather than a linear one and fit the power equations of:

$$AAOCN\text{-}6:\quad P[\%] = t^{0.40} \tag{1}$$

$$AAOCN\text{-}16:\quad P[\%] = 2.74 * t^{0.33} \tag{2}$$

Where $P[\%]$ is the permeance decrease in percentages and $t$ is the irradiation time in seconds. Both curves showed similar power dependence to irradiation time with AAOCN-16 being roughly 2.7 times higher due to higher amount of deposited pCN and smaller pores. The equations show that the amount of change reaches to a plateau as the irradiation time increases, especially in case of AAOCN-6 membrane. The irradiation time offers yet another level of control for the light switchable AAOCN membranes. Next, we tested the reversibility of the membranes via the light on-off cycles, each for 30 s and 60 s, for the AAOCN-6 membrane and observed a clear increase in the gas selectivities (~3% increase). In particular, we observed 2.8% and 2.9% increase in the $H_2/CO_2$ and $He/CO_2$ selectivities due to the large polarizability difference between the gas pairs and strong interactions with the charged pCN surface, respectively[46]. Considering the impact of irradiation time on the gas transport properties, we also measured the permeances of He, $H_2$, $N_2$, $O_2$, $CH_4$ and $CO_2$ at different irradiation times for AAOCN-8, -12 and -16 membranes (Supplementary Figs. 20–37). Expectedly, the reduction in He and $H_2$ gas permeances remained below 3-4% regardless of the irradiation time and the precursor amount for all AAOCN membranes (Supplementary Figs. 20–37). We observed a gradual

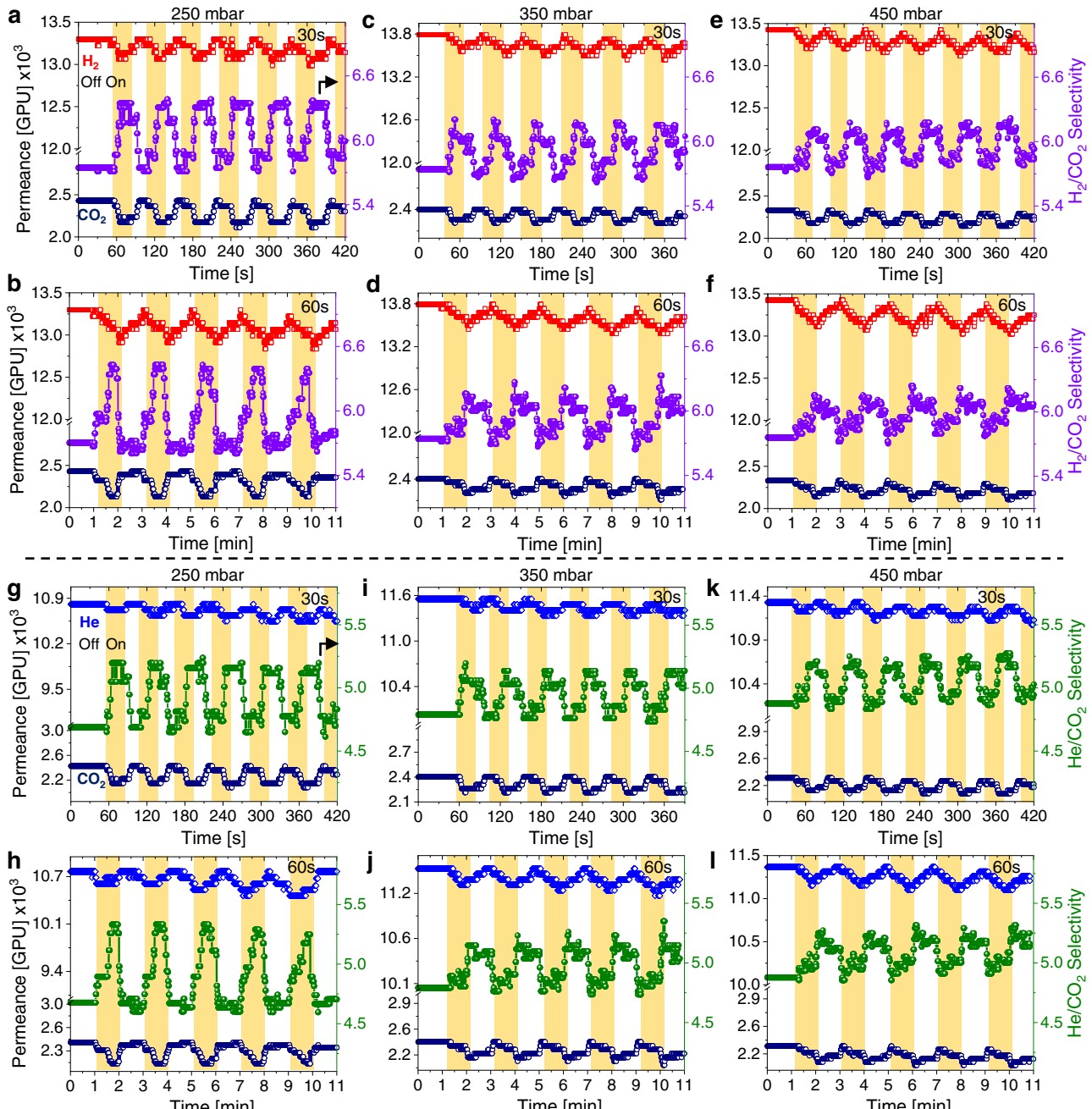

**Fig. 6 | Light induced reversible control of gas selectivity.** The change in gas permeance and $H_2/CO_2$ selectivity upon irradiation for 30 s (**a**) and 60 s (**b**) at 250 mbar, for 30 s (**c**) and 60 s (**d**) at 350 mbar, for 30 s (**e**) and 60 s (**f**) at 450 mbar, and the change in gas permeance and $He/CO_2$ selectivity upon irradiation for 30 s (**g**) and 60 s (**h**) at 250 mbar, for 30 s (**i**) and 60 s (**j**) at 350 mbar, for 30 s (**k**) and 60 s (**l**) at 450 mbar transmembrane pressures on **AAOCN-16** sample, respectively.

increase in the permeance change for $CO_2$ with respect to the precursor amount, which reached the highest $CO_2$ permeance decrease of 12% at 30 s and 15% at 60 s irradiation time, respectively, for the AAOCN-16 membrane at 250 mbar transmembrane pressure (Supplementary Figs. 38–39). These results were attributed to the low polarizabilities of He and $H_2$ compared to $CO_2$ and stronger interaction of the latter with the charged pCN surface, which was also verified by the DFT calculations results. The lower change in the $CH_4$ permeance compared to that of $CO_2$ despite of their similar polarizabilities (2.448 vs 2.507 Å$^3$) was attributed to the stronger interaction of $CO_2$ with the charged pCN surface compared to $CH_4$ though dipole-quadrupole interactions, which was also verified by DFT calculations[34].

We also investigated the correlation between the irradiation time and the amount of deposited pCN on the gas selectivities (Fig. 6 and Supplementary Fig. 40). We observed that higher amounts of pCN deposition and longer irradiation times led to higher gas/$CO_2$ selectivities. For instance, the $H_2/CO_2$ selectivities increased by 3.5% and 4% at an irradiation time of 30 s for AAOCN-8 and AAOCN-12 membranes at 250 mbar, respectively, and the same increase was recorded as 4 and 4.5% at irradiation time of 60 s (Fig. 5d and Supplementary Fig. 40a). Similarly, the $He/CO_2$ selectivity increased by 4 and 5% at 30 s and by 4.5 and 5.5% at 60 s irradiation times for the for AAOCN-8 and AAOCN-12 membranes at 250 mbar, respectively. The highest increase in the $H_2/CO_2$ and $He/CO_2$ selectivities were observed for AAOCN-16 sample, where the $H_2/CO_2$ and $He/CO_2$ selectivities increased by 12.5 and 12% at

30 s and by 15 and 14.5% at 60 s irradiation times at 250 mbar, respectively (Fig. 6). Moreover, we also tested longer irradiation time of 300 s for the AAOCN-16 membrane at 250 mbar and observed >22% increase in both $H_2/CO_2$ and $He/CO_2$ selectivities (Supplementary Fig. 41). It should be noted that the selectivity can be increased further by controlling either the light power or the irradiation time. Moreover, AAOCN membranes showed gas permeances in the range of $10^4$ GPU, which is very high compared to previously reported stimuli-responsive membranes (Supplementary Table 4). In addition, pCN-based light-switchable membranes also offer other advantages such as fast response time, tunability, single light source and a low energy input for switching (Supplementary Table 4).

In order to understand the relationship between the irradiation time and increase in the gas/$CO_2$ selectivity, we plotted the increase in the $H_2/CO_2$ and $He/CO_2$ selectivities upon irradiation on AAOCN-16 membrane vs irradiation time (Fig. 5c). Both gas mixtures showed a very similar logarithmic increase and were fitted with the following power functions:

$$H_2/CO_2 : \alpha[\%] = 4.57 * t^{0.28} \tag{3}$$

$$He/CO_2 : \alpha[\%] = 5.66 * t^{0.23} \tag{4}$$

where $\alpha$ [%] is the increase in the gas/$CO_2$ selectivity in percentages and $t$ is the irradiation time in seconds. Since the change in the $H_2$ and $He$ permeance was almost negligible upon light irradiation, the increase in the gas/$CO_2$ selectivity originated solely from the decreased $CO_2$ permeance at longer irradiation times owing to the higher polarizability and the stronger interactions of $CO_2$ with the charged pCN surface. For this reason, the gas/$CO_2$ selectivity vs light irradiation time curves have similar nature with the $CO_2$ permeance change vs irradiation time curve. Finally, the influence of different transmembrane pressures on gas selectivity were also tested at different irradiation times (Figs. 5d-e, 6 and Supplementary Figs. 22–40). The highest increase in the gas selectivities upon irradiation is observed at 250 mbar transmembrane pressure regardless of the irradiation time. As the increase in the transmembrane pressure led to a faster flow rate and lower gas-surface interactions, it resulted in a smaller change in the gas/$CO_2$ selectivities upon irradiation. For instance, the increase in the $H_2/CO_2$ and $He/CO_2$ selectivities dropped by ~1–2% at higher pressures on AAOCN-8 and AAOCN-12 membranes at all irradiation times (Fig. 5d-e and Supplementary Figs. 21–33, 40). However, the increase in the $H_2/CO_2$ and $He/CO_2$ selectivity for AAOCN-16 reduced from ~14.5 and 14.2% to 9.4 and 10.0%, respectively, when the pressure increased from 250 to 350 mbar at 60 s irradiation time (Figs. 5d, 6 and Supplementary Fig. 40a). On the other hand, the change in the $O_2/CO_2$, $N_2/CO_2$ and $CH_4/CO_2$ selectivities upon irradiation were found to be less dependent on the transmembrane pressure (Fig. 5e, and Supplementary Fig. 40b) owing to the high polarizabilities of these gases. These results clearly revealed that the extent of change in gas permeance and selectivity upon irradiation is governed by the polarizability of the gases as well as their interactions with the charged pCN surface. Moreover, these results also demonstrate the potential of pCN as light switchable membranes with a fast response time for the separation of gases with high polarizability differences. Nevertheless, application was not main focus of this study, these types of materials could potentially be useful for applications, in which quick changes in the membrane properties are required, i.e., natural gas purification[54,55] removal of large hydrocarbons from petroleum gas[56,57] and He purification[36].

## Discussion

In conclusion, we introduced polymeric carbon nitride as a light switchable gas separation membrane with a fast response time. Our results showed that the gas permeance across pCN membrane is fully governed by polarizability of the gases and their interaction with the charged pCN surface; while the permeance of the gases with low polarizabilities remained the same upon light irradiation, the permeance of the gases with higher polarizabilities changed significantly, enabling high gas selectivities for the binary gas mixtures with high polarizability differences. Moreover, light switchable pCN membranes offer multiple levels of control such as light power and irradiation time over the gas permeance and selectivity. These findings highlight the potential of pCN as light-switchable membranes for gas separation applications and open up new avenues for the development of responsive membranes based on physical transformations.

## Methods
### Materials
Commercial AAO membranes with 100 nm pores and 60 μm thickness were purchased from Sterlitech. Melamine was purchased from the Sigma-Aldrich and used without any purification. SEM images were obtained using ThermoFischer Scios 2 instrument using voltage of 3.0 kV and current of 0.40 nA. The samples were coated with 3.0 nm of gold on Cressington 208HR sputtering tool to avoid charging. FTIR spectra were obtained using Perkin-Elmer Frontier CSJ instrument in transmission mode. The XPS spectra were obtained on multi-purpose XPS, Sigma Probe, Thermo VG Scientific by using Monochromatic Al K(alpha) X-ray source. High resolution XPS spectra were deconvoluted using fityk software. The UV-Vis absorbance spectra of pCN membranes were obtained on Perkin Elmer Lambda 900 by using 2 nm/s scan speed. The emission spectra of the light source was measured on Avantes AvaSpec-2048 spectrometer.

### Polymeric carbon nitride deposition
Polymeric carbon nitride thin films were directly grown on porous AAO supports by using a low-pressure chemical vapor deposition (LPCVD) system, consisting of a 3 inch quartz tube with a dual-temperature zone furnace – with the first zone (upstream) at 300 °C and the second zone (downstream) at 550 °C, as previously described[23,38]. At these temperatures, melamine sublimes and undergoes through a thermal polymerization process. Ramp time and precursor amount were varied in order to obtain films of different thicknesses and composition (Supplementary Table 1). A silica boat loaded with melamine and AAO substrates in a vertical silica support were placed at the upstream and downstream zones, respectively. Before heating, the system was pumped down to 10 torr and flushed with $N_2$. Experiments were performed using a planarGROW-3 S-OS CVD System for Organic Semiconductors, provided by planarTECH. After the deposition, the system was cooled down to room temperature. The AAOCN samples were washed with acetone and dried before the gas permeation measurements.

### DFT calculations
The systems were first geometrically optimized. During geometry optimization calculations, the atoms located in the lower layer of the $g$-$C_3N_4$ slab were fixed at their bulk positions, while the coordinates of the atoms on the top layer were relaxed until forces drop below 0.03 eV Å$^{-1}$ threshold. The Brillouin zone was sampled using a $5 \times 5 \times 1$ Monkhorst–Pack k-point mesh[58]. The binding energy (EB) was computed as the difference between the energy of the adsorbed gas ($E_{g\text{-}C3N4+gas}$) and the sum of the energies of the free $g$-$C_3N_4$ surface ($E_{g\text{-}C3N4}$) and the corresponding gas-phase species ($E_{gas}$) according to following: $EB = (E_{g\text{-}C3N4+gas}) - (E_{g\text{-}C3N4} + E_{gas})$. The electronic exchange and correlation potential was modeled using the PBE-TS functional[59], which includes correction for weak dispersion interactions. During periodic DFT calculations ultrasoft pseudopotentials were employed with an energy cut-off of 550 eV. When solving the Kohn–Sham

equations, the electronic density was optimized until the associated energy reaches the threshold of $10^{-6}$ eV.

## Gas permeation measurements

The gas permeation measurements of blank AAO and AAOCN-X membranes were performed on a previously reported[44,50] custom-built system that consists of 3 mass flow controllers (Alicat Scientific), differential pressure controller (Alicat Scientific) and mass flow meters (Alicat Scientific) (Supplementary Fig. 42). Circular 13.0 mm sized (in diameter) membranes were positioned in a custom-designed holder made of transparent Plexiglas in between two O-rings (Vitton). The measurement system was flushed with the corresponding gas for extended duration before the measurements. Gas permeances were calculated from the slope of the flux (mol·s$^{-1}$·m$^{-2}$) vs transmembrane pressure (mbar) curve (Supplementary Fig. 8). The membranes were irradiated with 550 nm LED light source from the feed side and the light power intensity was controlled by controlling the current.

## Permeance calculation

Gas flow meters were set to read gas flow rates in unit of standard cubic centimeters per minute (sccm) and the pressure was set to millibar (mbar). Permeance was calculated from slope of the flow rate vs transmembrane pressure curve. Then, permeance was converted to SI units (mol·s$^{-1}$·m$^{-2}$·Pa$^{-1}$) using the following equation:

$$\frac{mol}{s} = \frac{1\,scm^3}{min} * \frac{1\,min}{60s} * \frac{1\,l}{1000\,scm^3} * \frac{1\,mol}{22.416\,l} \quad (5)$$

1 Gas Permeation Unit, GPU = $3.35 \times 10^{-10}$ mol·s$^{-1}$·m$^{-2}$·Pa$^{-1}$.

## Regression analysis

The measured results were processed using Origin Pro 2019b software. The number of iterations were run until the Chi-square tolerance of $1 \times 10^{-9}$ was reached. The fits were iterated until at least 95% of the data fitted the model ($R^2 > 0.95$).

## Data availability

The data that support the findings of this study are openly available in Zenodo at https://doi.org/10.5281/zenodo.7308537.

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

## Acknowledgements

Authors acknowledge the support from the Swiss National Science Foundation (SNF) for funding this research (200021-175947), in addition to the support from the Max Planck society.

## Author contributions

T.A. has performed all the membrane measurements. J.S.S. has pre-pared the membranes. M.Z. and A.O.Y. performed DFT calculations. A.C. and M.A. conceived and supervised the project, procured funds, and wrote the manuscript together with T.A and J.S.S.

## Competing interests

The authors declare no competing interests.
