## [Peer Review File · Nature Communications]

Fast light-switchable polymeric carbon nitride membranes for tunable gas separationREVIEWER COMMENTS

Reviewer #1 (Remarks to the Author):

In this manuscript, the authors used the method of chemical vapor deposition to obtain the light-switchable AAOCN-X membranes with various thickness and porosity. The light-switch ability of the membranes was proved by a series of tests on the gas permeance under different light power. In addition, some important parameters (such as transmembrane pressure and the irradiation duration) were considered to further verify the separation performance for the gas with high polarizabilities while offer multiple levels of control. The results showed that the gas permeance across AAOCN-X membrane is fully governed by polarizability of the gases and their interaction with the charged pCN surface.

Main concerns include:

1. As indicated in Introduction, light-switchable gas separation membranes are certainly interesting, but do they have any practical applications?
2. There is a schematic of the formation of a charged pCN surface upon light irradiation in Fig.1d, but there is no relevant characterization results about this point. Is there a feasible characterization method to characterize this model?
3. It's mentioned in lines 1 to 4 of page 4 that the position of the sample in the LPCD chamber has an enormous effect on the porosity of the membrane. What's the reason for this phenomenon?
4. Why is the gas/CO₂ selectivity of blank AAO membrane lower than Knudsen selectivity when the precursor amount is 0 in Fig.3a? This is not consistent with the sentence "Expectedly, the gas transport is governed by the Knudsen diffusion in the blank AAO membrane due to its large pores with a tubular shape" in line 17 on page 4.
5. As stated in the last sentence of "Since the highest gas selectivity was obtained at a maximum power level of 7.2 W/cm²..." on page 6. If we continue to increase the power level, what will happen for the gas selectivity?
6. As the sentence "Both curves showed similar power dependence to irradiation time with AAOCN-16 being roughly 2.7 times higher due to higher amount of deposited pCN and smaller pores." on page 7, can the constant "2.74" in Equation 2 be replaced as a function of the amount of deposited pCN?
7. It would look better if the author removed the scale lines of axes used in some figures in the paper (the upper axis of Fig.3a, the upper axis and the right axis of Fig.5a, d, e, etc.)

Reviewer #2 (Remarks to the Author):

The authors present a study on light-switchable poly-carbon nitrate (pCN) gas separation membranes with very fast response times of about 30-60s. This is thus far one of the fastest reported photoresponses in light-responsive materials for gas separation. The study contains tests on the influence irradiation power, irradiation time and amount of material deposited on the porous substrate. Gas permeation is measured extensively, also here the authors present measurements for different transmembrane pressures.

1) To your statement regarding electric field switching of membrane permeation "Recently, Knebel and coworkers reported an electric field-switchable metal-organic framework (MOF) membrane, zeolitic imidazolate framework, ZIF-8, which transforms into a polymorph structure under the influence of an electric field, thus resulting in a hindered gas transport. 8 However, the switching time of this membrane was on the order of tens of minutes (30 to 60 min) and a rather high voltage input of 500 V/mm was required." is not correct.

I read the paper and the switching towards the Cm polymorph was quite fast within seconds - minutes. When I see it correctly, the relaxation of the crystallographic phase back to I-43m took longer time, but only because no stimulus was applied to improve this phase change (e.g. T). The electric field that the samples are exposed to seem to be high for chemists, but it is rather low in reality and especially in the world of electrical engineering it is homeopathic. I would suggest rephrasing this part

a bit.

2) If you talk about long switching times in MOF-based light-switchable membranes (which is the biggest draw back of those materials), it would be important to also write down the numbers the switching times of the light-responsive MOFs, from Nat. Commun. (5 min to trans, 15 min to cis) and Chem. Mater. (about 2h) – there is a big difference whether guest molecules are switched or those incorporated into the backbone of the MOF.

3) The literature research for a paper on polymeric membrane should not only include reference towards MOF membranes. There are also examples for purely polymeric membranes with light switchable moieties and mixed matrix membranes:

DOI <https://doi.org/10.1039/C4SC02305F>

DOI <https://doi.org/10.1016/j.polymer.2021.124012>

DOI <https://doi.org/10.1016/j.eurpolymj.2019.05.051>

DOI <https://doi.org/10.1038/s41598-018-21263-7>

And more! Please do a complete literature review for yourself again to cover everything in this regard. I would suggest to work on the introduction to provide a broader look on already existing technologies with light-responsive materials.

4) It might be worth mentioning that isomeric photo responses are always happening on a very different time-scale than electron-hole separation as found in your material. Nevertheless, the photoresponse seems to be significantly longer than “seconds” as stated by you. In the gas permeation experiments it seems to be that you need half a minute to 1 minute to completely switch your materials. This is still fast, but compared to other photo-effects (e.g. photoluminescence which happens in ns) very long.

5) With increasing amount of precursor, your membrane gets thicker and at some point you don't have any more permeance. This is attributed to the pCN closing the pore-diameter of the underlying support correct? So how is it possible that you have such a dramatic switching effect when your pore-size is still about 150 nm? Can you explain the mechanism of the separation more and measure powders of your pCN polymers in adsorption under 550nm light to confirm your hypothesis?

6) How do you explain the gas permeation through the pCN. Usually, for amorphous polymers there is a solution-diffusion mechanism, whereas here you seem to form a gas-dense structure at a certain amount, when clogging the pores of the underlying substrate. Therefore, you have molecular sieving? Please explain the gas transport mechanism through pCN for me.

7) Overall, I think your methodology is completely supported and your data represents what you conclude. There is a high amount of novel measurements presented. Your permeation studies are extensive, and everything is covered, many parameters have been changed. I just don't understand how the separation performance is improved when pCN is switching. How strong are the charges you have in your material and how does it work in the Knudsen diffusion regime? How much is adsorbed in the pCN material or dissolved? Can you show some theoretical calculations on the gas adsorption mechanism?

8) You have measured several samples throughout your work, because this must be a reproducible result. In Figure 5 you could give error bars to represent how reproducible the results were.

Reviewer #3 (Remarks to the Author):

Ali Coskun and coworkers present a mesoporous membrane of polymeric carbon nitride on anodic aluminum oxide. The authors employ it for light-switchable separation of binary gas mixtures. They

find the gas permeation is governed by Knudsen diffusion through the mesopores of about 100nm diameter. Light irradiation was used to switch the selectivities. Changes of a few 10% were found. In comparison to other studies from the introduction (7-12), this is only a moderate switching effect. The work is based on their previous work(17), where such a carbon nitride membrane was used as light-driven ion pump.

The “ultrafast” light-switchable properties – see title – are not clearly elaborated.

Thus, my recommendation is that although it is an interesting research work, a more specified journal is more appropriate.

Before resubmission, the major issues must be corrected:

- The molecular structure of the polymeric carbon nitride needs to be presented.
- The issue of the “ultrafast” response is not explored enough. It seems that the fast response is reasoned from a small effect after short irradiation time. 30s in Supplementary Figure 35. Shouldn't the previous membranes, see introduction, give similar results? What is the improvement here? The fast response needs to be put on a solid data base, if it remains in the title.
- For the light-switchable separation of gas mixtures, the data is interpreted by the polarizability of the permeating gas. However, the data of CH₄ and CO₂ seems to contradict. The explanation and the cited references do not shed light into it. Please elaborate or correct “The lower change in the CH₄ permeance compared to that of CO₂ despite of their similar polarizabilities (2.448 vs 2.507) was attributed to the stronger interaction of CO₂ with the charged pCN surface compared to CH₄ though dipole-quadrupole interactions.²³”

Minor:

- The units in “...despite of their similar polarizabilities (2.448 vs 2.507)...” are missing.
- A few typos need to be corrected, like “.. transport trough the ...”,

Response to the Reviewer's Comments:

Reviewer 1:

Comment 1: In this manuscript, the authors used the method of chemical vapor deposition to obtain the light-switchable AAOCN-X membranes with various thickness and porosity. The light-switch ability of the membranes was proved by a series of tests on the gas permeance under different light power. In addition, some important parameters (such as transmembrane pressure and the irradiation duration) were considered to further verify the separation performance for the gas with high polarizabilities while offer multiple levels of control. The results showed that the gas permeance across AAOCN-X membrane is fully governed by polarizability of the gases and their interaction with the charged pCN surface.

Response: We would like to thank the reviewer for her/his positive evaluation of our work.

Comment 2: Main concerns include:

As indicated in Introduction, light-switchable gas separation membranes are certainly interesting, but do they have any practical applications?

Response: We thank the reviewer for mentioning this point. This field is relatively new and further developments are being made. Therefore, it is not easy at this stage to pinpoint exact application for these membranes. However, with higher and faster changes in the gas transport properties upon switching, light-switchable membranes could be used in places where dynamic changes in gas composition is present. We would like to, however, note that our main focus in this study is the fundamental understanding of light-switchable gas transport through polymeric carbon nitride membrane. In this regard, our study is more of a proof of concept and first demonstration of light-switchable gas separation using pCN membranes. The above-mentioned discussion was also implemented into the manuscript;

On pg 11: “. Nevertheless, these types of materials could.....”

Comment 3: There is a schematic of the formation of a charged pCN surface upon light irradiation in Fig.1d, but there is no relevant characterization results about this point. Is there a feasible characterization method to characterize this model?

Response: We thank the reviewer for this critical comment. In response to the reviewer's comment, we further detailed our mechanistic explanation and also added DFT calculations to

explain the changes in the gas permeability.

On pg 8: “The growth of pCN on the surface of the AAO leads to the formation of a heterojunction, in which the photoexcited electrons in the lowest unoccupied molecular orbital (LUMO) of pCN can migrate to the defect sites of AAO as it was also demonstrated by Li et al.⁵⁰ for g-C₃N₄/Al₂O₃ heterojunctions, where they observed an improved charge transfer. In order to understand the affinities of CO₂, CH₄, H₂, and He towards pCN, their binding energies were estimated by carrying out periodic plane-wave DFT calculations as implemented in the CASTEP program (version 19.11).⁵¹ In these periodic DFT calculations, we used a heptazine based graphitic carbon nitride (g-C₃N₄) structural model reported by Wang et.al,⁵² where the structure was determined through ab initio evolutionary search and consequent experimental verification. The systems were described using a simulation cell containing 2 layers of g-C₃N₄, obtained by replicating the unit cell 2 × 2, and 20 Å of vacuum, which was placed to avoid the interaction between two periodic images along the coordinate perpendicular to the g-C₃N₄ layers and a single gas molecule (Supplementary Fig. 17). Binding energies were calculated for both ground state and excited systems (Supplementary Table 3). For the latter, a core hole is assigned to one of the nitrogen atoms on the top layer; i.e. the layer exposed to the gas molecule, by exciting one of the 1s² electrons of the selected nitrogen atom, which resulted in an overall +1 charge on the top carbon nitride layer. As expected, in the ground state of g-C₃N₄ layers, we observed the highest binding energy for CO₂ followed by CH₄, which is also in agreement with the observed gas selectivities. When a core hole is created, we observed an increase in the dipole moment of g-C₃N₄ layers from 4.53 debye in the ground state to 4.95 debye in the charged state. Moreover, the calculation of binding energies of gases for the +1 charged state revealed that the binding energy of both CO₂ and CH₄ increased, which is expected owing to their higher polarizability, whereas rather modest changes observed for H₂ and He. These results explain the change in the gas selectivity of pCN membrane upon light irradiation and its correlation with the polarizability of gas molecules as it is expected that the higher the polarizability of the molecule the greater its diffusion will slow down near the pore surface due to increased electrostatic interactions after light irradiation.”

On pg 10: “.... surface, which was also verified by the DFT calculations results. The.....”
“dipole-quadrupole interactions, which was also verified by DFT calculations“.

On pg 13: “ **DFT calculations:** The systems were first geometrically optimized. During geometry optimization calculations, the atoms located in the lower layer of the g-C₃N₄ slab were fixed at their bulk positions, while the coordinates of the atoms on the top layer were relaxed until forces drop below 0.03 eV Å⁻¹ threshold. The Brillouin zone was sampled using a 5 × 5 × 1 Monkhorst–Pack k-point mesh.⁵⁷ The binding energy (EB) was computed as the

difference between the energy of the adsorbed gas ($E_{g-C_3N_4} + E_{gas}$) and the sum of the energies of the free g-C₃N₄ surface ($E_{g-C_3N_4}$) and the corresponding gas-phase species (E_{gas}) according to following: $EB = (E_{g-C_3N_4}) + (E_{gas}) - (E_{g-C_3N_4} + E_{gas})$. The electronic exchange and correlation potential was modelled using the PBE-TS functional,⁵⁸ which includes correction for weak dispersion interactions. During the periodic DFT calculations ultrasoft pseudopotentials were employed with an energy cut-off of 550 eV. When solving the Kohn–Sham equations, the electronic density was optimized until the associated energy reaches the threshold of 10^{-6} eV.

In the supporting information:

On pg 15: “

Supplementary Figure 17. A simulation box that illustrates the periodic DFT optimized position of CO₂ molecule over two layers of g-C₃N₄.”

On pg 32: Table S3. Binding energy of gases with pCN.

	Binding energy (kJ/mol)	
	pCN	pCN(+1)
CO ₂	-32.9	-35.5
CH ₄	-20.1	-22.9
H ₂	-10.8	-10.9
He	-3.2	-3.8

Comment 4: It's mentioned in lines 1 to 4 of page 4 that the position of the sample in the LPCD chamber has an enormous effect on the porosity of the membrane. What's the reason for this phenomenon?

Response: We thank the reviewer for this point. Thin film growth in a low-pressure chemical vapor deposition chamber is highly dependent on different process parameters that dictate the kinetics and thermodynamics in the reaction system. When considering the deposition on AAO membranes in a tubular reactor, one must investigate the best conditions to ensure a continuous growth both inside the AAO pores and the membrane surface without causing the pores to clog/produce voids within it – meaning the flow needs to be carefully controlled to ensure that the thin films deposit uniformly/with an optimized deposition rate. Under the right conditions, the reactant gas is transported to- and adsorbed by the heated AAO membrane, initiating the condensation process and interaction between migrating atoms (nucleation) on the membrane, resulting on the growth of thin films.

In a horizontal hot-wall tubular reactor, the flow of the reactants is generally in the viscous-dominated laminar regime – the molecules are slowed down close to the substrate, creating a stagnant layer (or boundary layer). In fact, for hot-wall tubular reactors, the flow is never always perfectly laminar throughout the flow stream, as a certain amount of turbulence always occurs. This turbulence changes the flow profile and causes vortex formations, increasing the thickness of the boundary layer, and thus also causing an uneven distribution/thickness of the thin films on the membranes. Placing the AAO membranes further away from the source and closer to the end of the flow stream caused what we have called in our original submission “pore blockage” – meaning the surface reaction on the membrane surface was too fast for the pores to be filled, covering them without filling, generating voids. This could have happened due to different factors as the vortex in the flow together with a depletion of reactant could result in gas-phase nucleation and higher randomness in the atoms diffusion mechanism – forming a wide range of clusters and islands on the top of the AAO arrays. However, more experiments would be needed to confirm exactly which factor is strongly influencing on the mentioned phenomenon.

We agree with the reviewer that the correlation between the deposition parameters and the thin film thickness/ membrane pores filling is an interesting aspect and have described the phenomenon in more detail in the revised version of our manuscript, as below:

On pg 5: “We hypothesize that this phenomenon is due to changes both in the reactant flow and temperature profile along the hot-wall tubular reactor of the low-pressure CVD system. When locating the substrate further away from the precursor, e.g., closer to the end of the furnace, gas-nucleation can occur, leading to local depletion of the reactant and non-homogeneous thickness. A change in the mobility-diffusion of the reactants on the surface of the AAO membrane can result in the overgrowth of neighboring grains, causing the pore surface to clog and voids within it.”

Comment 5: Why is the gas/CO₂ selectivity of blank AAO membrane lower than Knudsen selectivity when the precursor amount is 0 in Fig.3a? This is not consistent with the sentence “Expectedly, the gas transport is governed by the Knudsen diffusion in the blank AAO membrane due to its large pores with a tubular shape” in line 17 on page 4.

Response: We thank reviewer for bringing this point to our attention. AAO is a tubular inorganic membrane with an average pore size of ~130 nm with a thickness of 60 μm. This pore size is too large for molecular sieving and separation happens due to differences in the masses of gases (Knudsen diffusion). The reason why selectivity of AAO membranes is lower than Knudsen selectivity is due to its large pore size. When then pore size becomes smaller, the selectivity will approach Knudsen selectivity, which is observed in our case. At certain pore size if gas-pore wall interactions are strong enough, selectivity may surpass Knudsen selectivity due to surface diffusion. Nevertheless, to avoid confusion we revised the relevant section in the manuscript as shown below:

On pg 6: “Separation in blank AAO happens due to differences in the masses of gases (Knudsen diffusion) due to its large pores with a tubular shape (~60 μm).“

Comment 6: As stated in the last sentence of “Since the highest gas selectivity was obtained at a maximum power level of 7.2 W/cm²...” on page 6. If we continue to increase the power level, what will happen for the gas selectivity?

Response: We thank the reviewer for this point. The light power affects the gases with higher polarizabilities, therefore further increase in the power levels is expected to lead to a higher increase in the Gas/CO₂ selectivity. Maximum value that our system can reach is 7.2 W/cm²,

thus we were unable to test higher power levels.

The above mentioned was also incorporated into the text:

On pg 9: “It should be noted that increasing the light power level is expected to lead to a higher increase in the gas/CO₂ selectivity.”

Comment 7: As the sentence “Both curves showed similar power dependence to irradiation time with AAOCN-16 being roughly 2.7 times higher due to higher amount of deposited pCN and smaller pores.” on page 7, can the constant “2.74” in Equation 2 be replaced as a function of the amount of deposited pCN?

Response: We would like to thank the reviewer for mentioning this. Indeed, 2.74 is very close to differences in the masses of deposited pCN, $16/6=2.67$. However, our main point of comparing AAOCN-6 and AAOCN-16 was to highlight the effect of pore size. Since gas-pore wall surface interactions govern the amount of change upon irradiation, the critical parameter here is the pore size. Therefore, it is difficult to redefine this as a function of amount of deposited pCN. In response to the reviewer’s comment, we also tried the same fitting by keeping the power of time the same, however, the fits became poor and R^2 dropped below 0.90.

Comment 8: It would look better if the author removed the scale lines of axes used in some figures in the paper (the upper axis of Fig.3a, the upper axis and the right axis of Fig.5a, d, e, etc.)

Response: We thank the reviewer for these suggestions. We have corrected the figures per reviewer’s request:

On pg 22: “

Fig. 3 Gas permeation results of AAOCN membranes. (a) Average pore size, H_2/CO_2 and He/CO_2 selectivities vs the precursor amount. (b) The gas permeance vs precursor amount and molecular weight of the gas.”

On pg 24: “

Fig. 5 Light switchable gas transport behavior of AAOCN membranes at different irradiation times and transmembrane pressures. (a) The permeance decrease upon irradiation vs gas polarizability in AAOCN-X membranes at 250 mbar transmembrane pressure. (b) The CO₂ permeance decrease (%) upon irradiation vs irradiation time (s) on AAOCN-6 and AAOCN-16 membranes. (c) The increase in H₂/CO₂ and He/CO₂ selectivities upon irradiation vs the irradiation time on AAOCN-16 membrane at 250 mbar transmembrane pressure. (d) The increase in the gas selectivity over 60s irradiation at maximum light intensity of 7.2 W/cm² vs transmembrane pressure for H₂/CO₂, He/CO₂, H₂/CH₄ mixtures and (e) for CH₄/CO₂, N₂/CO₂, O₂/CO₂ on AAOCN-X membranes.”

Reviewer 2:

Comment 1: The authors present a study on light-switchable poly-carbon nitrate (pCN) gas separation membranes with very fast response times of about 30-60s. This is thus far one of the fastest reported photoresponses in light-responsive materials for gas separation. The study contains tests on the influence irradiation power, irradiation time and amount of material deposited on the porous substrate. Gas permeation is measured extensively, also here the authors present measurements for different transmembrane pressures.

Response: We would like to thank the reviewer for her/his positive evaluation of our work.

Comment 2: To your statement regarding electric field switching of membrane permeation “Recently, Knebel and coworkers reported an electric field-switchable metal-organic framework (MOF) membrane, zeolitic imidazolate framework, ZIF-8, which transforms into a polymorph structure under the influence of an electric field, thus resulting in a hindered gas transport.⁸ However, the switching time of this membrane was on the order of tens of minutes (30 to 60 min) and a rather high voltage input of 500 V/mm was required.” is not correct. I read the paper and the switching towards the Cm polymorph was quite fast within seconds - minutes. When I see it correctly, the relaxation of the crystallographic phase back to I-43m took longer time, but only because no stimulus was applied to improve this phase change (e.g. T). The electric field that the samples are exposed to seem to be high for chemists, but it is rather low in reality and especially in the world of electrical engineering it is homeopathic. I would suggest rephrasing this part a bit.

Response: We thank the reviewer for this suggestion. We have revised the stated parts according to reviewer’s suggestions:

On pg 2: “....gas transport.⁸ Switching of the membrane towards Cm polymorph was fast, however, the relaxation of polarized ZIF-8 required ~1.5 hours, which can be potentially improved by applying additional stimulus (e.g. temperature).⁸”

Comment 3: If you talk about long switching times in MOF-based light-switchable membranes (which is the biggest drawback of those materials), it would be important to also write down the numbers the switching times of the light-responsive MOFs, from Nat. Commun. (5 min to trans, 15 min to cis) and Chem. Mater. (about 2h) – there is a big difference whether guest molecules are switched or those incorporated into the backbone of the MOF.

Response: We thank the reviewer for this suggestion. We already showed the switching durations in the supporting information in Supplementary Table S3. Nevertheless, per reviewer’s suggestion we also added it into the manuscript. We have revised the stated parts according to reviewer’s suggestions:

On pg 3. “....from 14.7 to 10.1, however, the switching azobenzenes into *cis* and back to *trans* configurations required rather long irradiation times 120 and 106 min, respectively.²⁰ In another....”

On pg 3. “....Additionally, the switching times also improved to ~15 min for *cis* and ~5 min back to *trans*. These switching times are still....”

*Comment 4: The literature research for a paper on polymeric membrane should not only include reference towards MOF membranes. There are also examples for purely polymeric membranes with light switchable moieties and mixed matrix membranes:
DOI <https://doi.org/10.1039/C4SC02305F>
DOI <https://doi.org/10.1016/j.polymer.2021.124012>
DOI <https://doi.org/10.1016/j.eurpolymj.2019.05.051>
DOI <https://doi.org/10.1038/s41598-018-21263-7>*

And more! Please do a complete literature review for yourself again to cover everything in this regard. I would suggest to work on the introduction to provide a broader look on already existing technologies with light-responsive materials.

Response: We thank the reviewer for this suggestion. We have included the literature mentioned by the reviewer and also expanded the introduction further to include light-responsive membranes.

On pg 2, “....For instance, Zhu and coworkers showed tunable CO₂ uptake in porous organic polymers functionalized with azobenzene moieties, where after UV irradiation CO₂ uptake capacity increased up to 29%.¹² They explained the increase in CO₂ binding affinity with the difference in polarities of *cis* and *trans* azobenzene.¹² One of the easiest ways of preparing light-responsive membranes is to incorporate light-switchable moieties into a polymer matrix. First example of such photo-switchable membranes were used to study tunable ion transport.¹³ Later, Weh and coworkers embedded azobenzenes into the poly(methylmethacrylate) (PMMA) and observed a decrease in the permeance of CH₄, H₂, SF₆ etc. upon photoswitching.¹⁴ Other examples also include liquid crystal as host materials for azobenzenes,¹⁵ where switchable

nitrogen permeation was studied under constant pressure and volume.¹⁶ In another example, Bujak and coworkers synthesized azopolyimides and studied switchable permeation of He, N₂, O₂ and CO₂.¹⁷ They observed 1.64% change in the permeation of He, however, the CO₂ permeance decreased by 7.89% due to dipole-quadrupole interactions. Recently, Nocon-Szmajda copolymerized and molecularly dispersed azobenzenes in polyimide matrix and observed almost ~28% and ~22% decrease in CO₂/N₂ selectivity in the case of molecular dispersion and copolymerization, respectively, upon 5 min irradiation.¹⁸ However, the permeance values of these membranes were extremely low in the range of ~10⁻²-10⁻³ GPU. MOFs are also interesting materials for light-switchable gas separation applications as they can offer higher permeances. For instance, Prasetya and coworkers prepared light-responsive mixed-matrix membranes by using JUC-62 and PCN-250 as filler materials and matrimid as polymer matrix.¹⁹ They observed up to 8.5% decrease in CO₂ permeance upon irradiation at 15 wt.% JUC-62 loading, however, ~83 minutes of irradiation was required to observe this change. In another example, Knebel and coworkers incorporated azobenzene guest molecules.....”

Comment 5: It might be worth mentioning that isomeric photo responses are always happening on a very different time-scale than electron-hole separation as found in your material. Nevertheless, the photoresponse seems to be significantly longer than “seconds” as stated by you. In the gas permeation experiments it seems to be that you need half a minute to 1 minute to completely switch your materials. This is still fast, but compared to other photo-effects (e.g. photoluminescence which happens in ns) very long.

Response: We thank the reviewer for bringing up this point. We would like to clarify this point, the “ultrafast response” term was used to highlight the separation of electron and holes. The changes in the membrane start to take immediately within seconds after turning on the light. Nevertheless, to avoid confusion we have changed the title of the manuscript and replaced ultrafast with fast.

Above-mentioned changes were also embedded into the manuscript:

Title: “fast light-switchable polymeric carbon nitride membranes for tunable gas separation”

On pg 2: “...exhibited fast response....”

On pg 4: “Separation of electrons and holes happens on a very different timescale compared to molecular isomeric light response.”

On pg 11: “...such as fast response time”

On pg 11: “...with a fast response time”

On pg 12: “...with a fast response time”

***Comment 6:** With increasing amount of precursor, your membrane gets thicker and at some point you don't have any more permeance. This is attributed to the pCN closing the pore-diameter of the underlying support correct? So how is it possible that you have such a dramatic switching effect when your pore-size is still about 150 nm?*

Response: We thank the reviewer for bringing up this point. pCN is deposited both on the surface and inside the pores of the AAO support (Figure 2). Using more precursor leads to narrowing of the AAO pores due to pCN growth within the channels. The deposition of 20 gr of pCN led to an overdeposition on the surface of AAO support leading to pore blockage (Figure S3). However, using 16 gr of precursor results in a nearly uniform deposition of pCN both in the pores and on the surface leading to pores ranging from 5-50 nm with an average of 21.4 nm (Figure 2 and R1) The highest changes were observed for the AAOCN-16. When the pores are large, e.g. AAOCN-6 sample ~138 nm, the effect of light-switching is very low (>5%). The switching effect reaches to ~20% on AAOCN-16, which has ~21.4 nm pores on average, thus pointing to the importance of pore wall-gas interactions. Reducing the pore size further is expected to increase the switching effect due to an even stronger pore wall-gas interactions.

Figure R1. Schematic cross-sectional representation of gas transport through: a) blank AAO and b) AAOCN membranes. After deposition of pCN the pores are narrowed which leads to a stronger pore wall-gas interactions.

Comment 7: Can you explain the mechanism of the separation more and measure powders of your pCN polymers in adsorption under 550nm light to confirm your hypothesis?

Response: We thank the reviewer for bringing up this point. In response to the reviewer's comment, in order to explain the mechanism of separation, we performed DFT calculations as detailed below to calculate the binding energies of the gases with the neutral and charged pCN.

On pg 8: "The growth of pCN on the surface of the AAO leads to the formation of a heterojunction, in which the photoexcited electrons in the lowest unoccupied molecular orbital (LUMO) of pCN can migrate to the defect sites of AAO as it was also demonstrated by Li et al.⁵⁰ for g-C₃N₄/Al₂O₃ heterojunctions, where they observed an improved charge transfer. In order to understand the affinities of CO₂, CH₄, H₂, and He towards pCN, their binding energies were estimated by carrying out periodic plane-wave DFT calculations as implemented in the CASTEP program (version 19.11).⁵¹ In these periodic DFT calculations, we used a heptazine based graphitic carbon nitride (g-C₃N₄) structural model reported by Wang et.al,⁵² where the structure was determined through ab initio evolutionary search and consequent experimental verification. The systems were described using a simulation cell containing 2 layers of g-C₃N₄, obtained by replicating the unit cell 2 × 2, and 20 Å of vacuum, which was placed to avoid the

interaction between two periodic images along the coordinate perpendicular to the g-C₃N₄ layers and a single gas molecule (Supplementary Fig. 17). Binding energies were calculated for both ground state and excited systems (Supplementary Table 3). For the latter, a core hole is assigned to one of the nitrogen atoms on the top layer; i.e. the layer exposed to the gas molecule, by exciting one of the 1s² electrons of the selected nitrogen atom, which resulted in an overall +1 charge on the top carbon nitride layer. As expected, in the ground state of g-C₃N₄ layers, we observed the highest binding energy for CO₂ followed by CH₄, which is also in agreement with the observed gas selectivities. When a core hole is created, we observed an increase in the dipole moment of g-C₃N₄ layers from 4.53 debye in the ground state to 4.95 debye in the charged state. Moreover, the calculation of binding energies of gases for the +1 charged state revealed that the binding energy of both CO₂ and CH₄ increased, which is expected owing to their higher polarizability, whereas rather modest changes observed for H₂ and He. These results explain the change in the gas selectivity of pCN membrane upon light irradiation and its correlation with the polarizability of gas molecules as it is expected that the higher the polarizability of the molecule the greater its diffusion will slow down near the pore surface due to increased electrostatic interactions after light irradiation.”

On pg 10: “.... surface, which was also verified by the DFT calculations results. The.....”
“dipole-quadrupole interactions, which was also verified by DFT calculations“.

On pg 13: “ **DFT calculations:** The systems were first geometrically optimized. During geometry optimization calculations, the atoms located in the lower layer of the g-C₃N₄ slab were fixed at their bulk positions, while the coordinates of the atoms on the top layer were relaxed until forces drop below 0.03 eV Å⁻¹ threshold. The Brillouin zone was sampled using a 5 × 5 × 1 Monkhorst–Pack k-point mesh.⁵⁷ The binding energy (EB) was computed as the difference between the energy of the adsorbed gas (E_{g-C₃N₄} + E_{gas}) and the sum of the energies of the free g-C₃N₄ surface (E_{g-C₃N₄}) and the corresponding gas-phase species (E_{gas}) according to following: EB = (E_{g-C₃N₄}) + (E_{gas}) – (E_{g-C₃N₄} + E_{gas}). The electronic exchange and correlation potential was modelled using the PBE-TS functional,⁵⁸ which includes correction for weak dispersion interactions. During the periodic DFT calculations ultrasoft pseudopotentials were employed with an energy cut-off of 550 eV. When solving the Kohn–Sham equations, the electronic density was optimized until the associated energy reaches the threshold of 10⁻⁶ eV.

In the supporting information:

On pg 15: “

Supplementary Figure 17. A simulation box that illustrates the periodic DFT optimized position of CO₂ molecule over two layers of g-C₃N₄.”

On pg 32: Table S3. Binding energy of gases with pCN.

	Binding energy (kJ/mol)	
	pCN	pCN(+1)
CO ₂	-32.9	-35.5
CH ₄	-20.1	-22.9
H ₂	-10.8	-10.9
He	-3.2	-3.8

We also measured UV-Vis spectra of AAOCN-16 as requested by the reviewer. pCN showed a rather broad adsorption peak ranging from 250 to 500 nm, which is in good agreement with the previously reported spectra of pCN (Sunasee *et al. Environ Sci Pollut Res* **26**, 1082-1093), while blank AAO absorbs deep in UV region below 250 nm.

Figure R2. The UV-Vis spectra of AAOCN-16 and AAO blank.

Figure R3. The UV-Vis spectra of AAOCN-16 after subtraction of AAO blank spectra.

We also measured the emission spectra of our LED source, which showed two distinct peaks at ~450 and 550 nm at low power levels. Interestingly, increasing the power levels led to an increase in the intensity of the peak at 550 nm. These results indicate that extending the absorption band of pCN above 500 nm could lead to even stronger switching effect.

Figure R4. Light source emission spectrum at low power level.

Figure R5. Light source emission spectrum at high power level.

Above-mentioned changes were also included in the revised manuscript:

On pg 5: “We also measured the ultraviolet-visible (UV-Vis) spectra of AAOCN and blank AAO membranes (Supplementary Fig 4). pCN showed a rather broad adsorption peak ranging from 250 to 500 nm, which is in good agreement with the previously reported pCN,⁴³ while blank AAO absorbs deep in UV region below 250 nm. The UV-Vis absorption bands of pCN

also match with emission spectra of used LED light source (Supplementary Fig 5).”

On pg 13: “....The UV-Vis absorbance spectra of pCN membranes were obtained on Perkin Elmer Lambda 900 by using 2 nm/s scan speed. The emission spectra of the light source was measured on Avantes AvaSpec-2048 spectrometer.”

In the supporting information:

On pg 4:

Supplementary Figure 4. The UV-Vis spectra of (a) AAOCN-16 and AAO blank. (b) AAOCN-16 after subtraction of AAO blank spectra.

Supplementary Figure 5. The emission spectra of LED source (a) at low and (b) at high power levels.

Comment 8: How do you explain the gas permeation through the pCN. Usually, for amorphous polymers there is a solution-diffusion mechanism, whereas here you seem to form a gas-dense structure at a certain amount, when clogging the pores of the underlying substrate. Therefore, you have molecular sieving? Please explain the gas transport mechanism through pCN for me.

Response: We thank the reviewer for mentioning this point. As the tubular shaped pores of AAO serves as a structural template, the growth of pCN is expected to occur from the pore walls, thus leading to a pore narrowing with increasing pCN amount, which we also verified experimentally. As depicted in the Figure R1, the separation mechanism in blank AAO occurs via Knudsen diffusion. However, it transforms into surface diffusion after deposition of pCN. We revised the **Figure 1** in the main text to clarify this point.

Figure R1. Schematic cross-sectional representation of gas transport through: a) blank AAO and b) AAOCN membranes. After deposition of pCN the pores are narrowed which leads to a stronger pore wall-gas interactions.

On pg 20: “

Fig. 1 Schematic representations of pCN membrane preparation. Cross-sectional image of (a) blank AAO membrane (Inset: real picture of AAO membrane), (b) polymeric carbon nitride deposited AAOCN membrane (Inset: real picture of AAOCN membrane), (c) gas transport through AAOCN membrane without irradiation, (d) molecular structure of pCN (e) formation of a charged pCN surface upon light irradiation and its effect on gas transport through the membranes. (f) schematic side-view of formation of charges on pCN upon irradiation.

Comment 9: Overall, I think your methodology is completely supported and your data represents what you conclude. There is a high amount of novel measurements presented. Your permeation studies are extensive, and everything is covered, many parameters have been changed.

Response: We thank the reviewer for his/her positive evaluation of our work.

Comment 10: *I just don't understand how the separation performance is improved when pCN is switching. How strong are the charges you have in your material and how does it work in the Knudsen diffusion regime? How much is adsorbed in the pCN material or dissolved? Can you show some theoretical calculations on the gas adsorption mechanism?*

Response: We thank the reviewer for asking clarification of these points. As noted above, upon coating the surface of AAO with pCN we move from Knudsen to Surface diffusion regime. In order to further elaborate on the mechanism, we also performed DFT calculations as detailed below.

On pg 8: “The growth of pCN on the surface of the AAO leads to the formation of a heterojunction, in which the photoexcited electrons in the lowest unoccupied molecular orbital (LUMO) of pCN can migrate to the defect sites of AAO as it was also demonstrated by Li et al.⁵⁰ for g-C₃N₄/Al₂O₃ heterojunctions, where they observed an improved charge transfer. In order to understand the affinities of CO₂, CH₄, H₂, and He towards pCN, their binding energies were estimated by carrying out periodic plane-wave DFT calculations as implemented in the CASTEP program (version 19.11).⁵¹ In these periodic DFT calculations, we used a heptazine based graphitic carbon nitride (g-C₃N₄) structural model reported by Wang et.al,⁵² where the structure was determined through ab initio evolutionary search and consequent experimental verification. The systems were described using a simulation cell containing 2 layers of g-C₃N₄, obtained by replicating the unit cell 2 × 2, and 20 Å of vacuum, which was placed to avoid the interaction between two periodic images along the coordinate perpendicular to the g-C₃N₄ layers and a single gas molecule (Supplementary Fig. 17). Binding energies were calculated for both ground state and excited systems (Supplementary Table 3). For the latter, a core hole is assigned to one of the nitrogen atoms on the top layer; i.e. the layer exposed to the gas molecule, by exciting one of the 1s² electrons of the selected nitrogen atom, which resulted in an overall +1 charge on the top carbon nitride layer. As expected, in the ground state of g-C₃N₄ layers, we observed the highest binding energy for CO₂ followed by CH₄, which is also in agreement with the observed gas selectivities. When a core hole is created, we observed an increase in the dipole moment of g-C₃N₄ layers from 4.53 debye in the ground state to 4.95 debye in the charged state. Moreover, the calculation of binding energies of gases for the +1 charged state revealed that the binding energy of both CO₂ and CH₄ increased, which is expected owing to their higher polarizability, whereas rather modest changes observed for H₂ and He. These results explain the change in the gas selectivity of pCN membrane upon light irradiation and its correlation with the polarizability of gas molecules as it is expected that the higher the polarizability of the molecule the greater its diffusion will slow down near the pore

surface due to increased electrostatic interactions after light irradiation.”

On pg 10: “... surface, which was also verified by the DFT calculations results. The.....”
“dipole-quadrupole interactions, which was also verified by DFT calculations“.

On pg 13: “ **DFT calculations:** The systems were first geometrically optimized. During geometry optimization calculations, the atoms located in the lower layer of the g-C₃N₄ slab were fixed at their bulk positions, while the coordinates of the atoms on the top layer were relaxed until forces drop below 0.03 eV Å⁻¹ threshold. The Brillouin zone was sampled using a 5 × 5 × 1 Monkhorst–Pack k-point mesh.⁵⁷ The binding energy (EB) was computed as the difference between the energy of the adsorbed gas (E_{g-C₃N₄} + E_{gas}) and the sum of the energies of the free g-C₃N₄ surface (E_{g-C₃N₄}) and the corresponding gas-phase species (E_{gas}) according to following: EB = (E_{g-C₃N₄}) + (E_{gas}) – (E_{g-C₃N₄} + E_{gas}). The electronic exchange and correlation potential was modelled using the PBE-TS functional,⁵⁸ which includes correction for weak dispersion interactions. During the periodic DFT calculations ultrasoft pseudopotentials were employed with an energy cut-off of 550 eV. When solving the Kohn–Sham equations, the electronic density was optimized until the associated energy reaches the threshold of 10⁻⁶ eV.

In the supporting information:

On pg 15: “

Supplementary Figure 17. A simulation box that illustrates the periodic DFT optimized position of CO₂ molecule over two layers of g-C₃N₄.”

On pg 32: Table S3. Binding energy of gases with pCN.

	Binding energy (kJ/mol)	
	pCN	pCN(+1)
CO ₂	-32.9	-35.5
CH ₄	-20.1	-22.9
H ₂	-10.8	-10.9
He	-3.2	-3.8

Comment 11: You have measured several samples throughout your work, because this must be a reproducible result. In Figure 5 you could give error bars to represent how reproducible the results were.

Response: We thank the reviewer for this point. We have added errors bar to represent the reproducibility of results:

On pg 24: “

Fig. 5 Light switchable gas transport behavior of AAOCN membranes at different irradiation times and transmembrane pressures. (a) The permeance decrease upon irradiation vs gas polarizability in AAOCN-X membranes at 250 mbar transmembrane pressure. (b) The CO₂ permeance decrease (%) upon irradiation vs irradiation time (s) on AAOCN-6 and AAOCN-16 membranes. (c) The increase in H₂/CO₂ and He/CO₂ selectivities upon irradiation vs the irradiation time on AAOCN-16 membrane at 250 mbar transmembrane pressure. (d) The increase in the gas selectivity over 60s irradiation at maximum light intensity of 7.2 W/cm² vs transmembrane pressure for H₂/CO₂, He/CO₂, H₂/CH₄ mixtures and (e) for CH₄/CO₂, N₂/CO₂, O₂/CO₂ on AAOCN-X membranes.”

Reviewer 3:

Comment 1: Ali Coskun and coworkers present a mesoporous membrane of polymeric carbon nitride on anodic aluminum oxide. The authors employ it for light-switchable separation of binary gas mixtures. They find the gas permeation is governed by Knudsen diffusion through the mesopores of about 100nm diameter. Light irradiation was used to switch the selectivities. Changes of a few 10% were found. In comparison to other studies from the introduction (7-12), this is only a moderate switching effect. The work is based on their previous work(17), where such a carbon nitride membrane was used as light-driven ion pump. The “ultrafast” light-switchable properties – see title – are not clearly elaborated. Thus, my recommendation is that although it is an interesting research work, a more specified journal is more appropriate.

Response: We would like to thank the reviewer for her/his detailed evaluation of our work. While we agree that there are other reported membranes with higher switching effect, our approach offers much faster switching times. More importantly, our approach introduces a new switchable material along with a new switching mechanism to the field mainly saturated with azobenzene and spiropyrans. In fact, we would like to also note that our membranes can reach up to 20% change and we also outlined the parameters to control the switching effect in a systematic manner. As for the relationship with the previous work, while the switchability of pCN is demonstrated in the context of ion transport, this study represents the first application of pCN as a switchable gas separation membrane. Concerning the reviewer’s comment on the “ultrafast switchability”, we would like to clarify that we originally referred to the photoexcitation of pCN layer, however, we also understand the confusion this has caused. Accordingly, we removed the term “ultrafast” from the title as well as from the main text. We

would like to emphasize that the fast switchability of the pCN membrane has been verified experimentally and we also provided a detailed performance comparison with the previously reported switchable membranes, which clearly showed the superior performance of pCN membranes.

Above-mentioned changes were also embedded into the manuscript:

Title: “fast light-switchable polymeric carbon nitride membranes for tunable gas separation”

On pg 2: “...exhibited fast response....”

On pg 4: “Separation of electrons and holes happens on a very different timescale compared to molecular isomeric light response.”

On pg 11: “...such as fast response time”

On pg 12: “...with a fast response time”

On pg 12: “...with a fast response time”

Comment 2: Before resubmission, the major issues must be corrected:

- The molecular structure of the polymeric carbon nitride needs to be presented.

Response: We thank the reviewer for the input and have added a new paragraph on the molecular structure of polymeric carbon nitrides. Polymeric carbon nitrides are part of a class of binary compounds, named carbon nitrides – its final structure is dependent on synthesis/deposition parameters such as the temperature of condensation and carbon to nitrogen ratio. To characterize the molecular structure of pCN, we performed X-ray photoelectron spectroscopy along FTIR analysis. The XPS results revealed near ideal carbon content, however, nitrogen was lower than ideal value indicating presence of defects. We have added a short description on the introduction and added molecular structure of pCN in updated Figure 1 of the manuscript:

Figure R6. XPS survey spectra of AAOCN-16 sample. The XPS spectra revealed near ideal content of carbon and lower amount of nitrogen indicating presence of defects.

Figure R7. High resolution (a) C1s, (b) N1s and (c) O1s spectra of AAOCN-16 membrane. High resolution C1s and N1s spectra revealed typical -C-N-/C=N- moieties. Peaks observed in high resolution O1s spectra were attributed to AAO.

On pg 3: “Polymeric carbon nitride is part of a class of binary compounds, named carbon nitrides. It is synthesized from nitrogen-rich organic compounds such as urea and melamine through a thermal polymerization process.²⁶ Recent studies on the polymerization mechanism

that leads to the final 2D structure have shown that pCN is formed by linear linked heptazines connected by hydrogen bonds with an offset, e.g. the 2D sheets are not precisely aligned on the top of each other, which causes the molecule to bend.^{27,28} Thus, the structure of polymeric carbon nitride would strongly depend on deposition parameters and final carbon to nitrogen ratio (Figure 1).”

On pg 5: “Moreover, the molecular structure of pCN was analyzed using X-ray photoelectron spectroscopy (XPS). The XPS survey spectra revealed ~36% of carbon, ~50% of nitrogen and ~14% of oxygen content (Supplementary Fig. 2). High resolution C1s spectra on the other hand showed the presence of -C-C-/-C=C- and -C-N-/C=N- moieties at 285.5 and 288.9 eV, respectively (Supplementary Fig. 3a).^{23,26,41} Moreover, high resolution N1s spectra revealed the presence of peaks at 399.7 and 400.6 eV that were attributed to typical -C-N- and -C=N- bonds of carbon nitride, respectively (Supplementary Fig. 3b).^{23,26,41} Most importantly, the peaks observed on high resolution O1s spectra were attributed to AAO moieties and organic oxygen moieties were not detected (Supplementary Fig. 3c).⁴²”

On pg 13: “The XPS spectra were obtained on multi-purpose XPS, Sigma Probe, Thermo VG Scientific by using Monochromatic Al K(alpha)) X-ray source. High resolution XPS spectra were deconvoluted using fityk software. “

On pg 20:

“

Fig. 1 Schematic representations of pCN membrane preparation. Cross-sectional image of (a) blank AAO membrane (Inset: real picture of AAO membrane), (b) polymeric carbon nitride deposited AAOCN membrane (Inset: real picture of AAOCN membrane), (c) gas transport through AAOCN membrane without irradiation, (d) molecular structure of pCN (e) formation of a charged pCN surface upon light irradiation and its effect on gas transport through the membranes. (f) schematic side-view of formation of charges on pCN upon irradiation. “

In the supplementary information:

On pg 3:”

Supplementary Figure 2. XPS survey spectra of AAOCN-16 sample. The XPS spectra revealed near ideal content of carbon and lower amount of nitrogen indicating presence of defects.

Supplementary Figure 3. High resolution (a) C1s, (b) N1s and (c) O1s spectra of AAOCN-16 membrane. High resolution C1s and N1s spectra revealed typical -C-N-/C=N- moieties. Peaks observed in high resolution O1s spectra were attributed to AAO.”

Comment 3: - The issue of the “ultrafast” response is not explored enough. It seems that the fast response is reasoned from a small effect after short irradiation time. 30s in Supplementary Figure 35. Shouldn’t the previous membranes, see introduction, give similar results? What is

the improvement here? The fast response needs to be put on a solid data base, if it remains in the title.

Response: We thank the reviewer for this comment. We performed a thorough study of the literature of light-responsive membranes. Almost all of them are based on azobenzenes and rely on molecular switching. However, in our study, the switching arises from separation of holes and electrons which results in a charged pCN surface. This is completely different and happens on a very different timescale compared to molecular switching. Almost all of the azobenzene-based switchable membranes require at least 3-5 minutes of irradiation and the relaxation time is also slow. On the other hand, the switching of pCN occurs instantly after switching on the light and keeps increasing with the irradiation time. Additionally, the relaxation of our membranes are also very fast.

We also would like to clarify this point, the “ultrafast response” term was used to highlight the separation of electron and holes. The changes in the membrane start to take immediately within seconds after turning on the light. Nevertheless, to avoid confusion we have changed the title of the manuscript and replaced ultrafast with fast.

Above-mentioned changes were also embedded into the manuscript:

Title: “fast light-switchable polymeric carbon nitride membranes for tunable gas separation”

On pg 2: “...exhibited fast response....”

On pg 4: “Separation of electrons and holes happens on a very different timescale compared to molecular isomeric light response.”

On pg 11: “...such as fast response time”

On pg 12: “...with a fast response time”

On pg 12: “...with a fast response time”

Comment 4: - *For the light-switchable separation of gas mixtures, the data is interpreted by the polarizability of the permeating gas. However, the data of CH₄ and CO₂ seems to contradict. The explanation and the cited references do not shed light into it. Please elaborate or correct “The lower change in the CH₄ permeance compared to that of CO₂ despite of their similar polarizabilities (2.448 vs 2.507) was attributed to the stronger interaction of CO₂ with the charged pCN surface compared to CH₄ though dipole-quadrupole interactions.²³”*

Response: We thank the reviewer for this critical comment. In order to explain the mechanism

and interaction of gases with the pCN surface in its neutral and charged states, we performed DFT calculations as detailed below. The DFT calculations also verified higher binding affinity of CO₂ towards both neutral and charged pCN compared to CH₄, thus explaining the higher change in the permeance of CO₂.

On pg 8: “The growth of pCN on the surface of the AAO leads to the formation of a heterojunction, in which the photoexcited electrons in the lowest unoccupied molecular orbital (LUMO) of pCN can migrate to the defect sites of AAO as it was also demonstrated by Li et al.⁵⁰ for g-C₃N₄/Al₂O₃ heterojunctions, where they observed an improved charge transfer. In order to understand the affinities of CO₂, CH₄, H₂, and He towards pCN, their binding energies were estimated by carrying out periodic plane-wave DFT calculations as implemented in the CASTEP program (version 19.11).⁵¹ In these periodic DFT calculations, we used a heptazine based graphitic carbon nitride (g-C₃N₄) structural model reported by Wang et.al,⁵² where the structure was determined through ab initio evolutionary search and consequent experimental verification. The systems were described using a simulation cell containing 2 layers of g-C₃N₄, obtained by replicating the unit cell 2 × 2, and 20 Å of vacuum, which was placed to avoid the interaction between two periodic images along the coordinate perpendicular to the g-C₃N₄ layers, and a single gas molecule (Supplementary Fig. 17). Binding energies were calculated for both ground state and excited systems (Supplementary Table 3). For the latter, a core hole is assigned to one of the nitrogen atoms on the top layer; i.e. the layer exposed to the gas molecule, by exciting one of the 1s² electrons of the selected nitrogen atom, which resulted in an overall +1 charge on the top carbon nitride layer. As expected, in the ground state of g-C₃N₄ layers, we observed the highest binding energy for CO₂ followed by CH₄, which is also in agreement with the observed gas selectivities. When a core hole is created, we observed an increase in the dipole moment of g-C₃N₄ layers from 4.53 debye in the ground state to 4.95 debye in the charged state. Moreover, the calculation of binding energies of gases for the +1 charged state revealed that the binding energy of both CO₂ and CH₄ increased, which is expected owing to their higher polarizability, whereas rather modest changes observed for H₂ and He. These results explain the change in the gas selectivity of pCN membrane upon light irradiation and its correlation with the polarizability of gas molecules as it is expected that the higher the polarizability of the molecule the greater its diffusion will slow down near the pore surface due to increased electrostatic interactions after light irradiation.”

On pg 10: “.... surface, which was also verified by the DFT calculations results. The.....”
“dipole-quadrupole interactions, which was also verified by DFT calculations“.

On pg 13: “ **DFT calculations:** The systems were first geometrically optimized. During

geometry optimization calculations, the atoms located in the lower layer of the g-C₃N₄ slab were fixed at their bulk positions, while the coordinates of the atoms on the top layer were relaxed until forces drop below 0.03 eV Å⁻¹ threshold. The Brillouin zone was sampled using a 5 × 5 × 1 Monkhorst–Pack k-point mesh.⁵⁷ The binding energy (EB) was computed as the difference between the energy of the adsorbed gas (E_{g-C₃N₄} + E_{gas}) and the sum of the energies of the free g-C₃N₄ surface (E_{g-C₃N₄}) and the corresponding gas-phase species (E_{gas}) according to following: EB = (E_{g-C₃N₄}) + (E_{gas}) – (E_{g-C₃N₄} + E_{gas}). The electronic exchange and correlation potential was modelled using the PBE-TS functional,⁵⁸ which includes correction for weak dispersion interactions. During the periodic DFT calculations ultrasoft pseudopotentials were employed with an energy cut-off of 550 eV. When solving the Kohn–Sham equations, the electronic density was optimized until the associated energy reaches the threshold of 10⁻⁶ eV.

In the supporting information:

On pg 15: “

Supplementary Figure 17. A simulation box that illustrates the periodic DFT optimized position of CO₂ molecule over two layers of g-C₃N₄.”

On pg 32: Table S3. Binding energy of gases with pCN.

	Binding energy (kJ/mol)	
	pCN	pCN (+1)
CO ₂	-32.9	-35.5
CH ₄	-20.1	-22.9
H ₂	-10.8	-10.9
He	-3.2	-3.8

Comment 5: *Minor:*

- The units in “...despite of their similar polarizabilities (2.448 vs 2.507)...” are missing.

Response: We would like to thank the reviewer for bringing this point to our attention. The units of the polarizabilities are added.

The above discussion was also implemented into the manuscript:

On pg 10: “..... despite of their similar polarizabilities (2.448 vs 2.507 Å³) was attributed to the stronger interaction of CO₂ with the charged pCN surface compared to CH₄ though dipole-quadrupole interactions.³⁴”

Comment 6: - A few typos need to be corrected, like “.. transport trough the ...”,

Response: We would like to thank the reviewer for bringing this point to our attention. The typos have been corrected.

The above discussion was also implemented into the manuscript:

On pg 6:

“Gas transport through the pCN membranes: In order.....”

We would like to thank all the reviewers once again for their critical comments, which helped us to improve the quality of our manuscript further.

REVIEWER COMMENTS

Reviewer #1 (Remarks to the Author):

I am satisfied with response and revision to this submission, and the paper could be acceptable.

Reviewer #2 (Remarks to the Author):

As an answer to my comments, the authors changed the introduction accordingly. Further, they added modeling on the adsorption behavior of this material through charge carrier separation. They made their figures and figure captions more clear.

All my concerns have been erased. Thanks for the strong effort spent by the authors. I can now fully recommend publication of this study in Nature Communications.

Reviewer #3 (Remarks to the Author):

In the revised version, Ali Coskun and coworkers tried to answer all questions by all reviewers. From my point of view, the major part of the manuscript is scientifically sound, with only a few exceptions, see below, however, the impact and novelty of the present work is too limited for a high-ranking journal.

The material is not novel. The application of the material for light-powered membrane was already demonstrated by the authors in the liquid phase, now it is gas phase. Switching by light of the gas separation has also been presented. The switching ON-OFF performance of this material is below other published materials.

The authors try to highlight the fast response as unique selling point. However, a few statements are misleading:

In table S4, the response time and the selectivity change are compared with other materials. While for the literature references, the total irradiation time is used as response time, for their work, the authors use the estimated response time which is much shorter than the total irradiation time. That comparison seems biased.

On the other hand, when comparing the selectivity change, the value after the total irradiation time of 5min is used. Their value for H₂-CO₂ is 22%, which is significantly smaller than the state of the art, e.g. 160% in ref.7. and 300% in ref. 14.

Also, I can't see the benefit of a fast response when the total effect is minor.

A fair comparison would be the selectivity change per irradiation time, ideally considering the irradiation intensity too.

A further issue:

How is the response time determined? An estimation from Figure 5b/c, the exponential-fit time constant is about 10-50s, which is similar to other reported materials, like DOI: 10.1039/d2sc02405e. For equation 1 and 2, the logarithmic relation works only in a small time window. Infinite time does probably not result in an infinite effect. Does it? The authors should comment on this and possibly change their model.

Bottom line, my recommendation is to publish it after the revision in a more specified journal like Communications Chemistry or Scientific Reports.

Response to the Reviewer`s Comments:

Reviewer 1:

Comment 1: I am satisfied with response and revision to this submission, and the paper could be acceptable.

Response: We would like to thank the reviewer for her/his comments and suggestions that improved the quality of our work substantially.

Reviewer 2:

Comment 1: As an answer ty my comments, the authors changed the introduction accordingly. Further, they added modeling on the adsorption behavior of this material through charge carrier separation. They made their figures and figure captions more clear. All my concerns have been erased. Thanks for the strong effort spent by the authors. I can now fully recommend publication of this study in Nature Communications.

Response: We would like to thank the reviewer for her/his comments and suggestions that improved the quality of our work substantially.

Reviewer 3:

Comment 1: In the revised version, Ali Coskun and coworkers tried to answer all questions by all reviewers.

Response: We would like to thank the reviewer for her/his comments and suggestions that improved the quality of our work.

Comment 2: From my point of view, the major part of the manuscript is scientifically sound, with only a few exceptions, see below, however, the impact and novelty of the present work is too limited for a high-ranking journal.

Response: We thank the reviewer for the input. As we have already stated in the first round of revisions, this work is the first demonstration of light-switchable gas separation through polymeric carbon nitride with very fast switching times. Through our investigations, we also unveiled the underlying mechanism to achieve fast switching and to control the transport properties of gases. We are confident that these findings will have a significant impact in field, which is currently dominated by azobenzenes, and open up new research directions.

Comment 3: The material is not novel. The application of the material for light-powered membrane was already demonstrated by the authors in the liquid phase, now it is gas phase. Switching by light of the gas separation has also been presented. The switching ON-OFF performance of this material is below other published materials.

Response: We thank the reviewer for this comment. However, we are quite confused by the assessment of the reviewer, as we have already acknowledged in our original submission that light controlled ion transport has been demonstrated, but this is the first demonstration of a light switchable membrane based on polymeric carbon nitride. We would also like to note that aside from fast switching capability, the on-off response performance of our membrane is comparable with most of the earlier reported light-responsive materials as shown in Table S4. Our materials operate at very high permeance ranges ($\sim 10^4$ GPU) and yet we can still achieve more than 20% change. The better performing materials have very low permeance and operate on molecular regime, therefore the change in the selectivity is expected to be higher. While it is not the focus of the present study, it is indeed possible to further increase the change in the selectivity by sacrificing from the gas permeance. However, the biggest challenge in the field is to realize high selectivity at high permeance.

Comment 4: The authors try to highlight the fast response as unique selling point. However, a few statements are misleading:

In table S4, the response time and the selectivity change are compared with other materials. While for the literature references, the total irradiation time is used as response time, for their work, the authors use the estimated response time which is much shorter than the total irradiation time. That comparison seems biased.”

On the other hand, when comparing the selectivity change, the value after the total irradiation

time of 5min is used. Their value for H₂-CO₂ is 22%, which is significantly smaller than the state of the art, e.g. 160% in ref.7. and 300% in ref. 14.

Also, I can't see the benefit of a fast response when the total effect is minor.

A fair comparison would be the selectivity change per irradiation time, ideally considering the irradiation intensity too.

Response: We thank the reviewer for the suggestions. We already highlighted in the footnotes that the 22% change was obtained at the irradiation time of 5 minutes. Nevertheless, we have updated the table S4 in the supporting information to avoid the confusion. The major advantages of light-switchable carbon nitride are the fast response time and the magnitude of the response can be tuned by controlling the irradiation duration. We do agree with the reviewer that adding irradiation power values to table would be useful, however, we could not locate this information in most of the previous studies. As outlined in our response to the previous comment, the total effect is not minor and realized at very high permeance range. The references highlighted by the reviewer operate at very low permeance range and have long response times.

In-situ grown pCN on AAO	Redistribution of the charges on pCN	Light: 550 nm Irradiation time Irradiation power	~1 s**	H ₂ /CO ₂ : increased from 5.8 to 7.1 (22.4%) at 5 min of irradiation He/CO ₂ : increased from 4.4 to 5.4 (22.6%) at 5 min of irradiation H ₂ /CH ₄ : increased from ~2.8 to ~3.1 (10.7%) at irradiation of 1 minutes	H ₂ : 13298 He: 10792	This work
--------------------------------------	--	--------	--	---	------------------

* 1 GPU = 3.35x10⁻¹⁰ mol s⁻¹ m⁻² Pa⁻¹

** The response time is the time required for the change to be observed. The amount of change is directly proportional with the irradiation duration.

The above-mentioned discussion is also included in the supporting information of the manuscript:

On pg 35:

In-situ grown pCN on AAO	Redistributio n of the charges on pCN	Light: 550 nm Irradiation time Irradiation power	~1 s**	H ₂ /CO ₂ : increased from 5.8 to 7.1 (22.4%) at 5 min of irradiation He/CO ₂ : increased from 4.4 to 5.4 (22.6%) at 5 min of irradiation H ₂ /CH ₄ : increased from ~2.8 to ~3.1 (10.7%) at irradiation of 1 minutes	H ₂ : 13298 He: 10792	This work
--	--	--	--------	--	---	----------------------

* 1 GPU = $3.35 \times 10^{-10} \text{ mol s}^{-1} \text{ m}^{-2} \text{ Pa}^{-1}$

** The response time is the time required for the change to be observed. The amount of change is directly proportional with the irradiation duration.

Comment 5: *A further issue:*

How is the response time determined? An estimation from Figure 5b/c, the exponential-fit time constant is about 10-50s, which is similar to other reported materials, like DOI: 10.1039/d2sc02405e.

Response: We would like to thank the reviewer for this comment. The response time is determined as the time when the changes start to occur. For conventional materials, such as azobenzenes, the changes occur gradually with the increasing irradiation time owing to the isomerization of azobenzene moieties. However, for our material, the change occurs instantly and the longer irradiation times leads to a further increase in the selectivity.

As for the study referred by the reviewer, it is not a gas separation membrane, but features azobenzene moieties trapped within the pores of a metal organic framework. Expectedly, 100 seconds were required for the free azobenzene moieties to switch and to reach the steady-state, which contains a mixture of cis/trans azobenzene moieties. However, the relaxation time of these materials was substantially longer. It is also not clear whether the same switching times can be achieved under dynamic gas separation conditions.

Comment 6: *For equation 1 and 2, the logarithmic relation works only in a small time window. Infinite time does probably not result in an infinite effect. Does it? The authors should comment on this and possibly change their model.*

Response: We would like to thank the reviewer for bringing this point to our attention. The equations 1 and 2 are power equations with $n < 1$, indicating that the effect becomes less at

higher irradiation durations. This is true especially for the equation 1 which is modeled for AAOCN-6 membrane. The amount of change reaches to a plateau and the further increase in the irradiation duration will result in a less change. In this sense, the model nicely reflects this point. In order to clarify this point, we added the following sentence in the manuscript:

On pg 10:

“...pores. The equations show that the amount of change reaches to a plateau as the irradiation time increases, especially in case of AAOCN-6 membrane. The....“

Comment 7: Bottom line, my recommendation is to publish it after the revision in a more specified journal like Communications Chemistry or Scientific Reports.

Response: We thank the reviewer for his/her feedback and as highlighted in our responses, this study will make a significant impact in the field currently saturated by azobenzenes and also open up new directions for the further development of light switchable gas separation membranes.